# Assembly of the infant gut microbiome and resistome are linked to bacterial strains in mother's milk

Pamela Ferretti [1,9], Mattea Allert[2,9], Kelsey E. Johnson[2,3], Marco Rossi[1], Timothy Heisel[4], Sara Gonia[4], Dan Knights[5,6], David A. Fields[7], Frank W. Albert[2], Ellen W. Demerath[8], Cheryl A. Gale [4] & Ran Blekhman[1] ✉

The establishment of the gut microbiome in early life is critical for healthy infant development. Although human milk is recommended as sole nutrition for the infant, little is known about how variation in the milk microbiome shapes the microbial communities in the infant gut. Here, we quantified the similarity between the maternal milk and the infant gut microbiomes using 507 metagenomic samples collected from 195 mother-infant pairs at one, three, and six months postpartum. Microbial taxonomic overlap between milk and the infant gut was driven by *Bifidobacterium longum*, and infant microbiomes dominated by *B. longum* showed greater temporal stability than those dominated by other species. We identified numerous instances of strain sharing between milk and the infant gut, involving both commensal (e.g. *B. longum*) and pathobiont species (e.g. *K. pneumoniae*). Shared strains also included typically oral species such as *S. salivarius* and *V. parvula*, suggesting possible transmission from the infant's oral cavity to the mother's milk. At one month, the infant gut microbiome was enriched in biosynthetic pathways, suggesting that early colonisers might be more metabolically independent than those present at six months. Lastly, we observed significant overlap in antimicrobial resistance gene carriage within mother-infant pairs. Together, our results suggest that the human milk microbiome has an important role in the assembly, composition, and stability of the infant gut microbiome.

The gut microbiome composition and maturation in early life plays an important role in the development of the immune system[1,2], nutrient absorption[3,4], and metabolism regulation[5]. Its assembly is influenced by microbes acquired from the mother[6–9], among other sources[10], starting from the first day of life[11]. While the impact of the maternal gastrointestinal, vaginal, oral and cutaneous microbial species on the infant gut microbiome assembly and development have been extensively investigated[6,11–15], the role of the maternal breast milk microbiome in modulating infant gut microbes remains poorly understood.

[1]Section of Genetic Medicine, Department of Medicine, University of Chicago, Chicago, IL, USA. [2]Department of Genetics, Cell Biology, and Development, University of Minnesota, Minneapolis, MN, USA. [3]Center for Genetic Epidemiology, Keck School of Medicine of the University of Southern California, Los Angeles, CA, USA. [4]Department of Pediatrics, University of Minnesota, Minneapolis, MN, USA. [5]Department of Computer Science and Engineering, University of Minnesota, Minneapolis, MN, USA. [6]BioTechnology Institute, College of Biological Sciences, University of Minnesota, Minneapolis, MN, USA. [7]Department of Pediatrics, the University of Oklahoma Health Sciences Center, Oklahoma City, OK, USA. [8]Division of Epidemiology and Community Health, University of Minnesota School of Public Health, Minneapolis, MN, USA. [9]These authors contributed equally: Pamela Ferretti, Mattea Allert. ✉e-mail: blekhman@uchicago.edu

Human breast milk represents, ideally, the sole source of nutrition for the infant in the first six months of life[16]. Nevertheless, milk remains heavily understudied, estimated to be less than 0.2% of all human-associated public metagenomic samples, based on data from 2021[17]. Maternal milk provides the infant not only with nutrients (macro-nutrients, vitamins, minerals) but also bioactive components, including human milk oligosaccharides, HMOs, maternal immune cells, antibodies, and live bacteria[18–20]. The milk, and the bacteria that it carries, have therefore the potential to significantly impact infant gut microbiome composition, stability, and functionality. In addition, the presence of antimicrobial resistance genes in the milk microbiome can further influence the infant's health and its gut microbiome composition[21]. The microbiome could also underlie the benefits of exclusive breastfeeding for a variety of chronic conditions including asthma[22–24], childhood obesity[25], type 1 diabetes[26], and allergic disease[27].

Despite its clear importance for infant health, research on the human milk microbiome has lagged behind studies of other human body sites. This is partly due to the nature of the sample itself: rich in human cells and fat content, and characterized by low microbial biomass[28], milk samples are challenging to process and sequence using standard approaches[28]. This is especially true for colostrum, the milk produced in the first days postpartum, as its particularly low production volume further exacerbates the low biomass challenge typical of all milk samples[29], and its collection would reduce the amount available for the infant. The few available milk microbiome cohorts are either limited by a small sample size or by lack of paired infant samples. Studies have also been hindered by the use of 16S rRNA gene sequencing[30,31], which cannot reliably identify microbial strains[32] and does not provide insights into the functional potential of the bacterial communities. Previous amplicon studies have identified a "core" microbial signature of human breast milk microbiome, mostly dominated by taxa belonging to the *Staphylococcus* and *Streptococcus* genera, followed by genera such as *Lactobacillus*, *Bifidobacterium*, *Veillonella*, and *Escherichia* among others[19,33]. Bifidobacteria are of particular interest due to their critical role in infant health[34]. They are among the most abundant taxa in the gut of breastfed infants, and their depletion has been associated with immune and metabolic disorders[34–36]. Bifidobacteria, in particular *B. longum subsp. infantis*, are able to digest HMOs[37]. The ability to use HMOs as a carbon source, as well as the genetic mechanisms underlying this ability, varies across members of the *Bifidobacterium* genus[38]. Notably, the depletion of genes required for the degradation of HMOs in bifidobacteria has been linked to systemic inflammation and immune dysregulation in infants, highlighting the importance of bifidobacteria-mediated HMOs degradation in infant health[39].

In this work, we investigate the relationship between the maternal milk microbiome and the infant gut microbiome by analyzing paired breast milk and infant stool samples collected during the first six months postpartum from a large cohort of healthy, predominantly breastfeeding mothers and their term infant. Using high-throughput shotgun metagenomics, we characterize microbial species composition and temporal stability, and identify multiple instances of strain-level sharing between mothers and infants, as well as strain persistence in the infant gut over time. We then analyze the microbial functional potential, with particular focus on the antimicrobial resistance gene carriage of both breast milk and infant stool samples over time.

## Results

### *Bifidobacterium longum* drives the limited compositional overlap between the infant gut and the maternal milk microbiome

We collected and sequenced 507 microbiome samples from 195 mother-infant pairs recruited in Minneapolis, USA. All infants were born at term and were exclusively breastfed at one month of age. Breast milk was collected at one and three months postpartum ($n = 173$), while infant stool samples were collected at one and six months postpartum ($n = 334$, Fig. 1A). The majority of infants were vaginally born (75%), exclusively breastfed at six months (76%), and antibiotics naive (67% had not received antibiotics by six months of age) (Supplementary Fig. 1 and Supplementary Data 1, 2). Species-level taxonomic profiling was performed with MetaPhlAn4[40] (Supplementary Data 3).

Maternal milk was characterized by a significantly lower species richness than infant stool samples ($p = 2.1 \times 10^{-16}$ when comparing milk at one month versus infant stool at one month, and $p = 1.7 \times 10^{-5}$ when comparing milk at three months versus infant stool at one month, paired t-test; Fig. 1B). Microbial diversity in both infant stool samples and maternal milk moderately increased over time ($p = 0.06$, and $p = 0.09$, respectively; paired t-test). The milk microbiome was dominated by *Bifidobacterium longum* (Fig. 1C). Other prominent members of the milk microbiome included *Bifidobacterium breve* and *Bifidobacterium bifidum*, skin-associated species such as *Staphylococcus epidermidis* and *Cutibacterium acnes*, and oral-associated species such as *Streptococcus salivarius* (Fig. 1C). This is consistent with previous studies showing detecting skin-associated taxa in milk in addition to bifidobacteria[41,42]. Maternal secretor status was not significantly associated with differences in the milk microbiome alpha and beta diversity ($p = 0.76$ paired t-test; Supplementary Fig. 2A, B).

The infant gut microbiome at one month was dominated by *B. longum*, *B. breve*, *B. bifidum*, *Escherichia coli*, *Bacteroides fragilis*, *Phocaeiola vulgatus*, and *Phocaeiola dorei*, as well as by species typically associated with the oral cavity, such as *Veillonella parvula*, *Veillonella atypica*, and *Veillonella dispar* (Fig. 1D). *B. longum* was the most prevalent species found in both milk and infant stool samples (55.2% and 98.2%, respectively, Fig. 1C, D and Supplementary Fig. 3). Other taxa found in both body sites included *B. breve*, *B. bifidum*, *B. fragilis*, and *P. vulgatus* (Fig. 1C, D). *Bifidobacterium* was the most prevalent and abundant genus in milk and stool samples at six months (Supplementary Fig. 4A). At the family taxonomic level, the prevalence of Bifidobacteriaceae increased over time (Supplementary Fig. 4B). Based on the dominant (most abundant) species, the infant stool samples could be divided in four groups, hereafter referred to as pre-dominance groups: the first three dominated by *B. longum*, *B. breve*, and *B. bifidum*, respectively, and a fourth group dominated by species not belonging to the *Bifidobacterium* genus, mostly *E. coli*, *B. fragilis*, and *P. vulgatus* (Fig. 1D). 29.3% of infant stool samples were dominated by *B. longum*, 13.8% by *B. breve*, 5.4% by *B. bifidum* and 51.5% by non-bifidobacteria species (Supplementary Fig. 5). Similarly, 35% of milk samples were dominated by *B. longum*, while 44.5% were dominated by non-bifidobacteria species.

While beta diversity analysis showed that milk and infant stool microbiome composition was overall distinct (Fig. 1E, F), both body sites were dominated by *B. longum* (Fig. 1C, D), indicating that *B. longum* is the main driver of the limited compositional overlap between a typical milk microbiome and a typical infant gut microbiome at the population level. In both milk and infant stool samples, *B. longum* subsp. *longum* (*BL. longum*) and *B. longum* subsp. *infantis* (*BL. infantis*) frequently co-existed in the same sample (81.4% of milk and stool samples, 52.1% of milk samples at one month, 83.4% and 95.6% of stool samples at one and six months, respectively; Supplementary Fig. 6A, B). *BL. infantis* was present at significantly higher relative abundance than *BL. longum* in milk samples at one and three months postpartum ($p = 3.2 \times 10^{-2}$ and $p = 4 \times 10^{-2}$, respectively; t-test) and in infant stool samples at six months of age ($p = 1.5 \times 10^{-6}$; t-test - Supplementary Fig. 6C). *BL. longum* was significantly enriched in infant stool samples at one month of age as compared to *BL. infantis* ($p = 1.6 \times 10^{-3}$; t-test - Supplementary Fig. 6C).

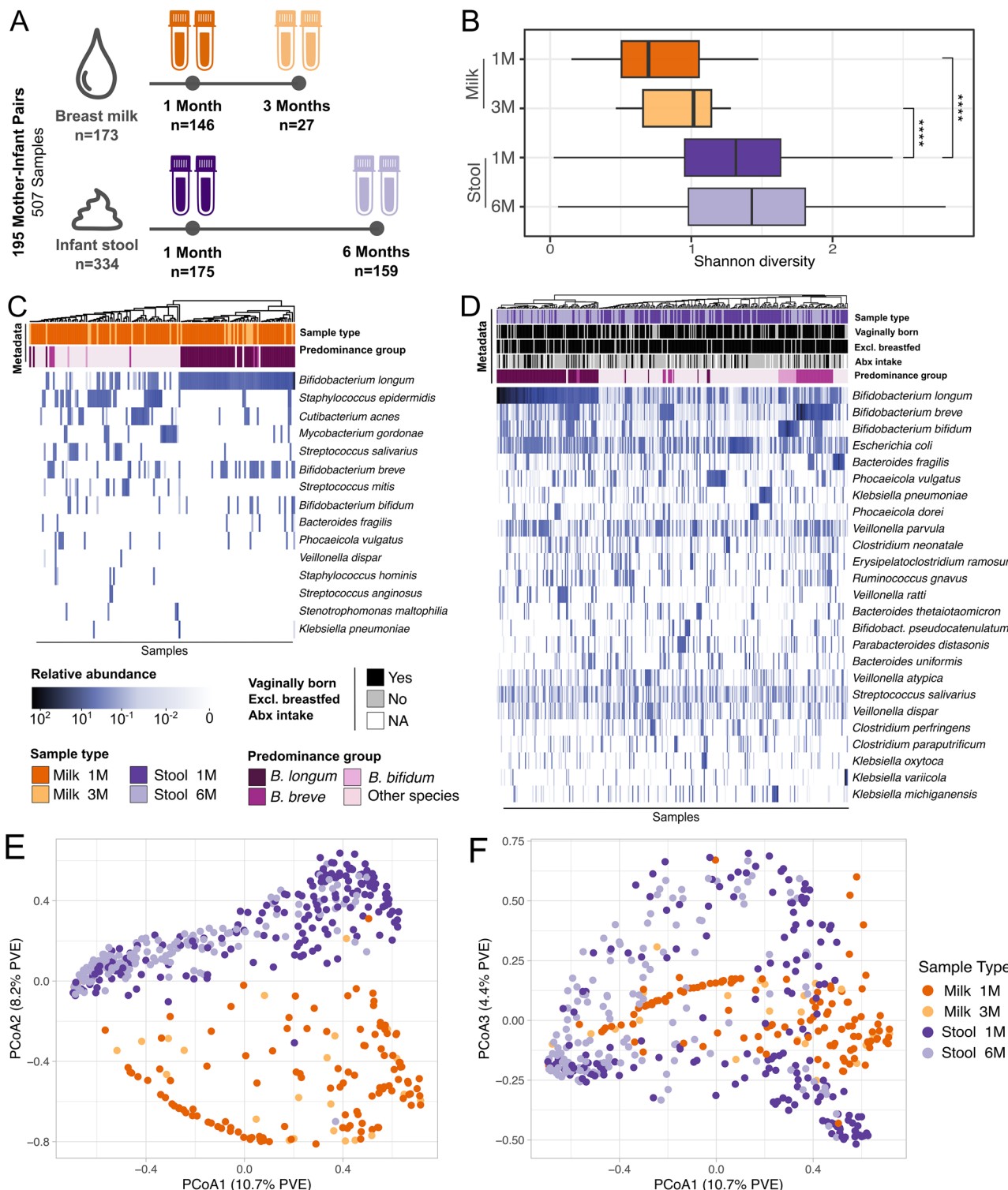

**Fig. 1 | Study design, microbial diversity, and taxonomic composition of maternal milk and infant stool microbiomes over time. A** Study design overview for the 507 samples collected from 195 mother-infant pairs. Infant stool samples (*n* = 334) were collected at one (1 M) and six months (6 M) of life. Maternal breast milk samples (*n* = 173) were collected one and three months (3 M) after delivery. **B** Shannon diversity distribution for infant stool and maternal milk samples over time. *P*-values calculated using two-sided paired *t*-test (* for *p* ≤ 0.05, ** for *p* ≤ 0.01 and **** for *p* ≤ 0.0001). Samples included: 122 for milk at one month, 25 for milk at three months, 175 for stool one month and 159 for stool at six months of age. Taxonomic composition of (**C**) the most prevalent and abundant species found in the human breast milk and (**D**) in the infant gut microbiome samples in relation to sample collection time point, predominance group and other relevant infant metadata. The predominance group identifies the most abundant species in each sample. Ordination plot based on Bray-Curtis distance between samples for PCoA1 and PCoA2 (**E**) and PCoA1 and PCoA3 (**F**), colored by body site of origin and sampling time. Percentage of variance explained (PVE) is reported in parentheses.

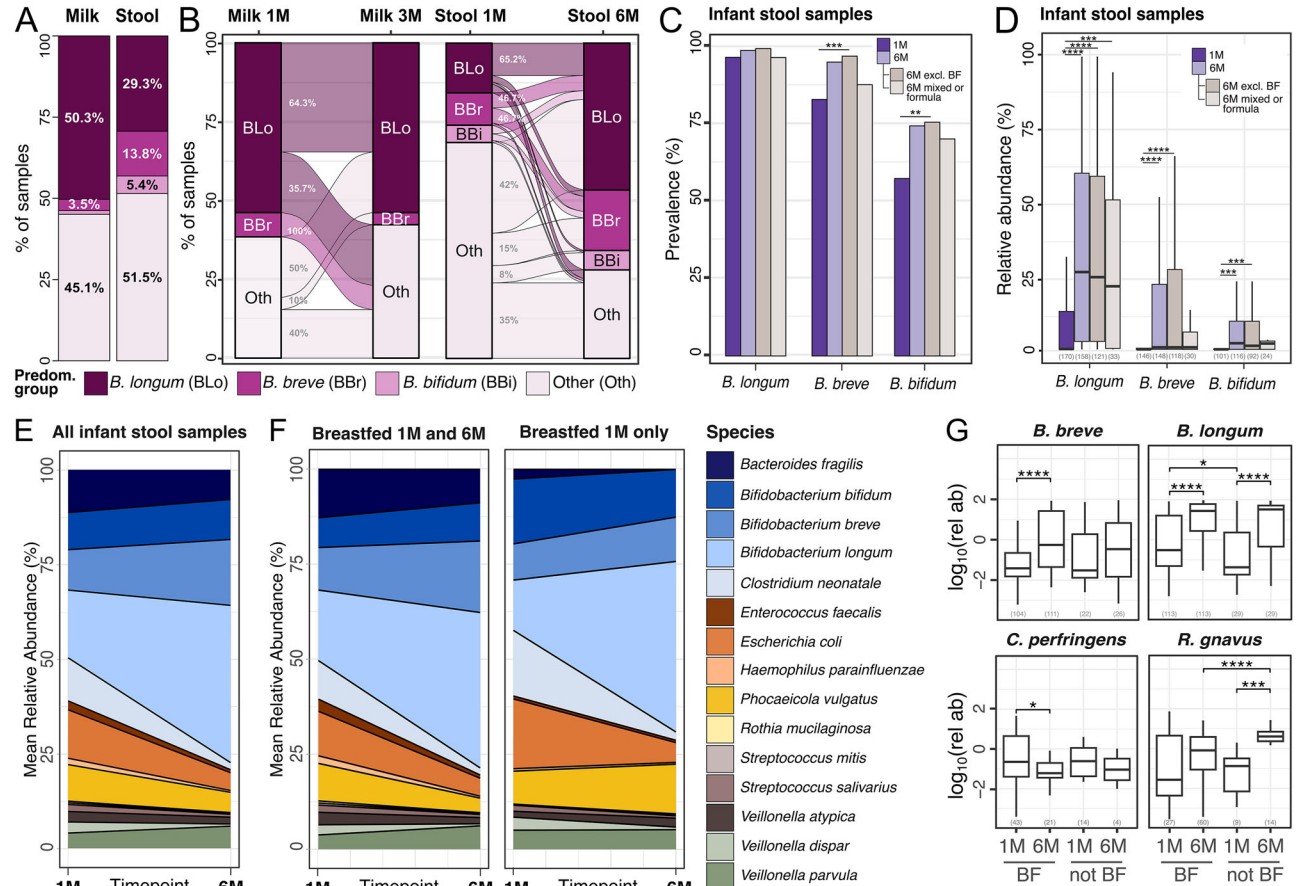

**Fig. 2 | Dynamics of *Bifidobacterium* predominance, abundance, and persistence in the infant gut and maternal milk microbiomes, stratified by breastfeeding status. A** Prevalence of each predominance group in milk and infant stool samples and **B** the transition of samples between predominance groups over time. Each sample is assigned one of four predominance groups indicated by the different colors. Numbers of samples included: $n = 145$ for milk at one month, $n = 25$ for milk at three months, $n = 175$ for stool samples at one month and $n = 159$ for stool samples at six months of age. BLo = *B. longum*, BBr = *B. breve*, BBi = *B. bifidum*. **C** Prevalence of *Bifidobacterium* species at one and six months of age. Prevalence at six months is further stratified by exclusive breastfeeding status ($n = 121$ for exclusive breastfeeding; $n = 34$ for mixed/formula diet). Samples lacking breastfeeding information at six months were excluded from the stratified subgroups ($n = 4$). Bonferroni-adjusted $p$-values were calculated using two-sided Fisher's exact

tests. **D** Relative abundance of *Bifidobacterium* species in infants at one and six months of age. Relative abundances at six months of age are further stratified by exclusive breastfeeding status. Relative abundances were calculated only on samples in which the species was present. Sample size is reported in brackets. *P*-values calculated using two-sided Wilcoxon rank sum, * for $p \leq 0.05$, **$p \leq 0.01$, *** for $p \leq 0.001$ and **** for $p \leq 0.0001$. Reported $p$-values are adjusted using Bonferroni method. **E** Species persistence in the infant gut across all samples and **F** stratified by breastfeeding at six months. See Supplementary Fig. 7 for plots with separation between *BL. infantis* and *BL. longum*. **G** Relative abundance of some of the differentially abundant species between one and six months when divided by breastfeeding (BF) at six months. BF exclusively breastfed. Sample size is reported in brackets. *P*-values calculated with two-sided paired *t*-test and adjusted with BH correction. **** for $P \leq 0.0001$, *** for $P \leq 0.001$, ** for $P \leq 0.01$ and * for $P \leq 0.05$.

## *B. longum* in the infant gut becomes less prevalent over time and is associated with a more stable microbiome

To investigate the stability of the microbiome in our samples, we first focused on the predominance groups described in the previous section. Among bifidobacteria, *B. longum* was the species with the highest prevalence in both breast milk (55.2%) and infant stool samples (98.2%, Fig. 2A). Specifically, 50.3% of milk samples were dominated by *B. longum* (17.3% by *BL. infantis* and 28.3% by *BL. longum*), 3.5% by *B. breve*, and 1.1% by *B. bifidum* (Fig. 2A and Supplementary Fig. 7A). Across the *Bifidobacterium* genus, *B. longum* was also the most abundant, present with a mean relative abundance of 12.4% in milk samples and 21.9% in infant stool samples, Fig. 2A, B). Mothers whose breast milk was dominated by *B. longum* (in particular *BL. infantis*) exhibited the most stable milk microbiome over time. Specifically, 64.3% of milk samples in which *B. longum* was the predominant species at one month postpartum maintained this dominance at three months (Fig. 2B and Supplementary Fig. 7B). In contrast, none of the milk samples initially dominated by *B. breve* retained that predominance at the three-month

time point (Fig. 2B). In infant stool samples, the percentage of samples dominated by *B. longum* increased over time, from 15.7% at one month to 46.6% at six months of age (Fig. 2B). Conversely, the percentage of samples dominated by non-bifidobacteria decreased from 68.5% to 28.1% (Fig. 2B).

In the infant, *B. longum* was found in 97.7% of the stool samples collected at 1 month of age (85.6% for *BL. infantis* and 88.5% for *BL. longum*, Fig. 2C and Supplementary Fig. 7C), while *B. breve* and *B. bifidum* were found in 83.9% and 58% of the samples in this timepoint, respectively (Fig. 2C). Overall, the prevalence of bifidobacteria in the infant gut increased from one to six months of age, particularly for those infants that were still exclusively breastfed at six months (Bonferroni adjusted $p = 0.08$ for *B. longum*, $p = 2.4 \times 10^{-4}$ for *B. breve*, and $p = 1.8 \times 10^{-3}$ for *B. bifidum*, Fisher's exact test; Fig. 2C). A similar trend was observed in *B. longum* subspecies (Bonferroni adjusted $p = 3.9 \times 10^{-3}$ for *BL. longum*, $p = 1.4 \times 10^{-2}$ for *BL. infantis*, Fisher's exact test; Supplementary Fig. 7C). Consistent with the prevalence patterns, we observed significant temporal increases in the relative abundance

of several *Bifidobacterium* species in the infant gut. *B. longum* mean relative abundance significantly increased from one to six months of age (Bonferroni adjusted $p = 6.2 \times 10^{-12}$, Wilcoxon Rank Sum; Fig. 2D). Subspecies analysis showed that *BL. infantis* increased from 3.2% mean relative abundance at one month to 23.8% at six months (Bonferroni adjusted $p = 3.6 \times 10^{-17}$, Wilcoxon Rank Sum) and that *BL. longum* increased from 8.7% mean relative abundance at one month to 9.1% at six months (Bonferroni adjusted $p = 1.1 \times 10^{-5}$, Wilcoxon Rank Sum) (Supplementary Fig. 7D). Overall, *BL. longum* was more abundant than *BL. infantis* at one month but not at six months of age (Bonferroni adjusted $p = 4.1 \times 10^{-11}$, Wilcoxon Rank Sum, Supplementary Fig. 7D). The trend of increasing mean relative abundance from one to six months of age was consistently observed across other *Bifidobacterium* species, regardless of breastfeeding practices at six months (Bonferroni-adjusted $p = 2.4 \times 10^{-6}$ for *B. breve* and $p = 1.6 \times 10^{-4}$ for *B. bifidum*, Wilcoxon rank-sum test; Fig. 2D), albeit with substantial inter-individual variability (Supplementary Fig. 8).

Next, we looked at the longitudinal stability of the infant gut microbiome composition at the species-level. We found that *B. longum* (in particular *B. infantis*) was more abundant at six months compared to one month (BH-adjusted $p = 2.2 \times 10^{-10}$, *t*-test), while *E. coli* and *V. dispar* decreased over time (BH-adjusted $p = 5.3 \times 10^{-3}$ and $p = 6.7 \times 10^{-3}$, respectively, *t*-test; Fig. 2E and Supplementary Fig. 7E). We then investigated if there were species that were differentially abundant between the two timepoints when stratifying the infants for exclusive breastfeeding. Infants that were exclusively breastfed through six months of age showed a significant increase in *B. longum* and *B. breve* relative abundances (BH-adjusted $p = 3.6 \times 10^{-9}$, $p = 2.2 \times 10^{-5}$, respectively, paired *t*-test; Fig. 2F, G and Supplementary Fig. 9A), and a significant reduction in *C. perfringens* (BH-adjusted $p = 1.3 \times 10^{-2}$, paired *t*-test; Fig. 2G), compared to those who were not exclusively breastfed at six months of age. Infants that were exclusively breastfed at one month but not at six months showed an increase in *R. gnavus* (BH-adjusted $p = 9.7 \times 10^{-4}$, *t*-test; Fig. 2G) as well as *B. longum* (BH-adjusted $p = 1.6 \times 10^{-4}$, *t*-test; Fig. 2F, G) and both its subspecies (BH-adjusted $p = 2.5 \times 10^{-5}$ for *BL. infantis* and BH-adjusted $p = 2.5 \times 10^{-2}$ for *BL. longum*, *t*-test; Supplementary Fig. 9A). We then compared the relative abundance of taxa in the gut of infants introduced to solid foods and those exclusively breastfed at six months of age. We found no taxa with statistically significant differential abundance between infants who were exclusively breastfed at six months and those who were introduced to solid foods (FDR > 0.05). Overall, the infant stool samples that were dominated by *B. longum* at both one and six months of age showed the most stable community composition over time (Supplementary Fig. 9B).

## Strains shared between breast milk and the infant gut include bifidobacteria, pathobionts, and taxa typically found in the gut and oral microbiome

While maternal microbial transmission is well documented for other body sites, it remains unclear whether specific taxa are shared between breast milk and the infant gut, which microbes are involved, and how frequently such sharing occurs. To investigate these questions, we quantified species- and strain-level overlap between breast milk and infant stool samples to investigate potential vertical transmission events (see Methods). We found that species sharing between a mother's milk and her infant's stool samples decreased over time. On average, 10.1% of taxa found in infant stool samples at one month of age were also found in their respective mother's milk collected at one month postpartum (Fig. 3A). This percentage significantly decreased when considering infant stool samples at 6 months of age (7.2%, $p = 1.5 \times 10^{-2}$ *t*-test; Fig. 3A). The most commonly shared species between milk and infant stool samples at one month postpartum was *B. longum* (shared on average in 44.3% of mother-infant pairs – 46.9% for *BL. infantis* and 41.7% for *BL. longum*), followed by *S. epidermidis*

(38.7%), *B. breve* (26%), and *S. salivarius* (18%, Fig. 3B). When considering stool samples at one and six months combined, *BL. infantis* was shared in 54.4% of the mother-infant pairs, while *BL. longum* was shared in 47.2% of pairs (Fig. 3C). *B. breve* and *B. bifidum* were characterized by a significantly lower rate of sharing (30% and 11%, respectively, Fig. 3C). While a positive correlation between the relative abundance of bifidobacteria in milk and the corresponding infant stool samples was observed in certain mother-infant pairs, no significant association was detected at the population level (Supplementary Fig. 10).

While informative of the potential microbial sharing between mothers and infants, species-level taxonomic profiling is not sufficient to identify transmission events, due to the genetic variability between conspecific strains[32]. We therefore performed strain-level profiling with StrainPhlAn4[40], which allowed us to reliably identify strains that are found in pairs of samples, such as a mother and her infant, defined as strain sharing. We reconstructed a total of 77 strains, of which 65 were found only in infant fecal samples and 12 were found in milk samples, due to the lower coverage. We identified twelve instances of strain-sharing between a mother's milk and her infant's stools, spanning across six mother-infant pairs and ten different taxa: *B. longum* (SGB17248), *B. bifidum* (SGB17256), *P. vulgatus* (SGB1814), *E. coli* (SGB10068), *K. pneumoniae* (SGB10115), *K. variicola* (SGB10114), *V. parvula* (SGB6939), *R. mucilaginosa* (SGB16985), *S. salivarius* (SGB8005), and a strain belonging to the *Streptococcus* genus (SGB8007_group, Fig. 3D and Supplementary 4). In particular, species like *R. mucilaginosa* and *S. salivarius* are considered typical members of the infant oral microbiome, while *P. vulgatus* and *K. pneumonia* and bifidobacteria are more commonly found in the infant gut[11]. All strains identified in milk samples were from samples collected at one month postpartum. Strains shared between milk and infant stool samples at one month of age were also found in stool samples collected at six months of age in 25% of cases (Fig. 3E). Overall, strain sharing between milk and infant stool samples was higher at one month of age compared to samples collected at six months of age (Fig. 3E), while no strain-sharing event was detected in milk samples collected at three months postpartum, likely due to the low microbial yield in these samples.

In addition to mother-infant sharing, strain analysis can also provide information on strain persistence in the infant gut over the first six months of age. We found that 19.4% of the strains identified in infant stool samples at one month persisted in the infant stools until six months of age (Supplementary Data 4). The taxa most commonly found to be persistent in the infant gut included *E. coli* (23.2% of persistence events), *B. longum* (13.4%), *B. fragilis* (8.5%), *B. vulgatus* (8.5%), and *B. bifidum* (8.5%, Supplementary Data 4 and Fig. 3F). Vaginally delivered infants had a higher number of persistent strains than infants that were born via C-section ($p = 1.8 \times 10^{-2}$, *t*-test; Fig. 3G). Exclusive breastfeeding at six months of age, antibiotic intake, and maternal secretor status were not significantly associated with differences in strain persistence in the infant gut ($p = 0.08$, $p = 0.25$, and $p = 0.84$, respectively).

## The maternal milk and early infant gut microbiomes are enriched in metabolic pathways for the biosynthesis of essential amino acids

Next, we investigated the functional potential of the maternal milk and infant gut microbiomes using HUMAnN3[43], a method that profiles microbial metabolic pathways from metagenomic data. The most prevalent pathways identified in the infant stool samples were associated with de-novo biosynthesis of molecules, in particular of essential amino acids (such as valine, isoleucine, threonine, and lysine) and ribonucleotides (Fig. 4A). Other abundant pathways were associated with cell structure (i.e. peptidoglycan maturation), and energy metabolism. The pathways involved in peptidoglycan maturation were

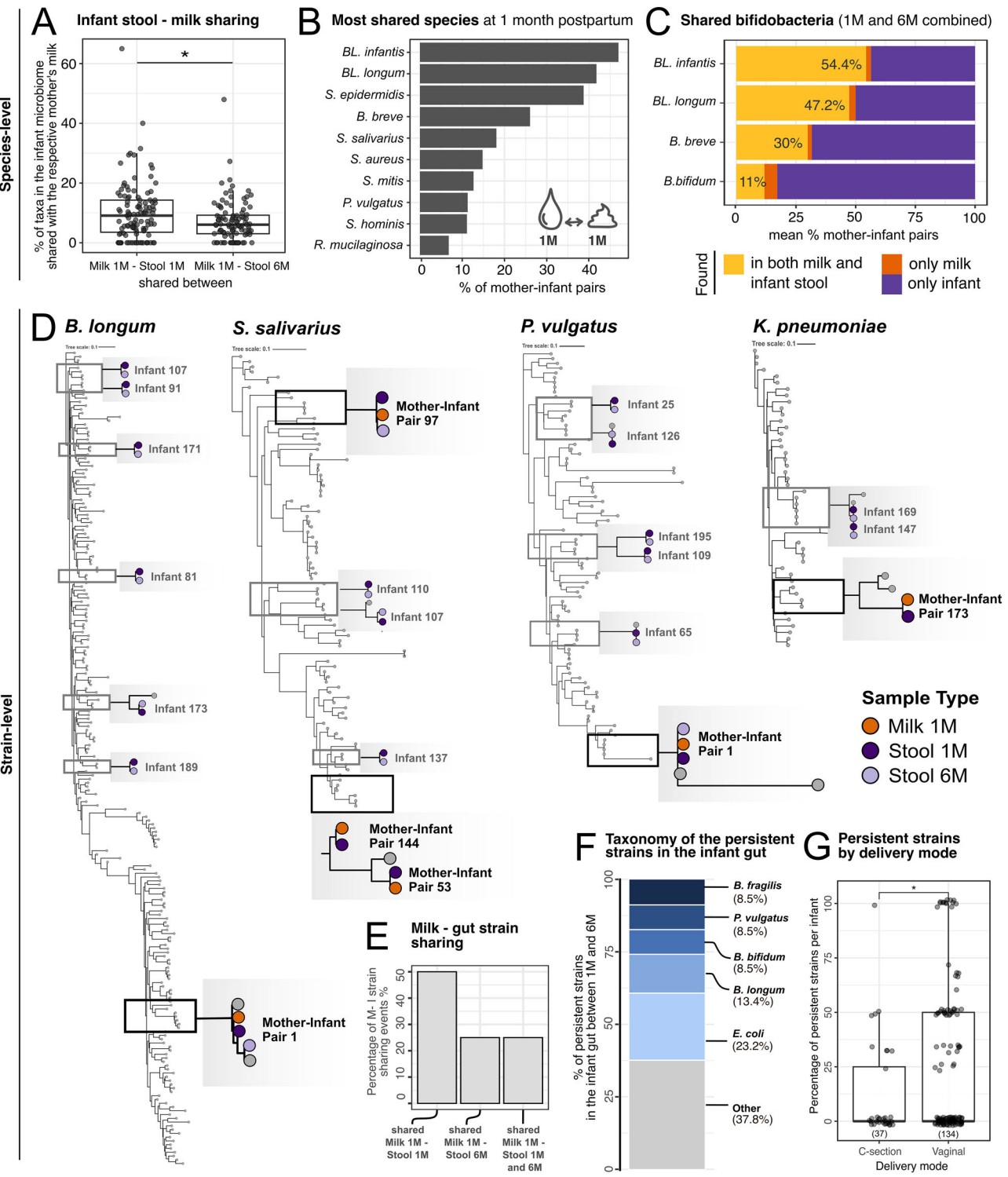

present at higher relative abundances in infant stool samples at one month compared to six months ($p = 2.7 \times 10^{-15}$, *t*-test; Fig. 4A), and this increase was particularly prominent for stool samples at one month dominated by non-bifidobacteria species (Supplementary Fig. 11A). In milk, the most prevalent and abundant pathways were associated with the biosynthesis of nucleotides and amino acids (such as valine and isoleucine), cell metabolism (sucrose biosynthesis, glycogen degradation, pyruvate fermentation and folate transformations), and cell structure (peptidoglycan maturation) (Fig. 4B). In infant stool samples, the pathways associated with essential amino acid biosynthesis were significantly more abundant at one month compared to six months,

and this trend was significantly more pronounced for infants that were no longer exclusively breastfed at six months (Bonferroni-adjusted $p = 5.2 \times 10^{-4}$ for infants exclusively breastfed at one and six months, and $p = 4.6 \times 10^{-12}$, for infants exclusively breastfed at one month only, *t*-test; Fig. 4C). In particular, milk samples at one month dominated by *B. breve*, were associated with the highest abundance of metabolic pathways associated with the biosynthesis of essential amino acids (Supplementary Fig. 11B). In milk, pathways associated with the biosynthesis of essential amino acids were significantly more abundant at three months compared to one month postpartum (Bonferroni-adjusted $p = 2.4 \times 10^{-2}$, *t*-test; Fig. 4C)

**Fig. 3 | Species and strain-level sharing between maternal milk and infant stool microbiomes, and temporal persistence of microbial strains in the infant gut.** **A** Percentage of the infant stool microbiome that is shared with the respective mother, across all mother-infant pairs. Species sharing between milk samples at one month postpartum and infant stool samples at one month and six months of age are shown on the left and right boxplot, respectively. Two-sided *t*-test, * for $p \leq 0.05$. Only mother-infant pairs for which milk at one month and stool samples at one and six months were available were included ($n = 108$ pairs). **B** Overview of the ten most frequently shared species between the infant stool samples at one month of age and the respective mother's milk samples at one month postpartum. *BL. infantis* = *B. longum* subsp. *infantis*; *BL. longum* = *B. longum* subsp. *longum*. **C** Mean percentage of *Bifidobacterium* species that were shared between milk and infant stool samples, or found in only one of the body sites, across all mother-infant pairs. Sharing was assessed between milk at one month and infant stool at one and six

months, and milk at three months and infant stool at six months. Only pairs with at least one milk and one stool sample were included. **D** Examples of strain sharing between milk and infant stool samples for *B. longum* (SGB17248), *Phocaeicola vulgatus* (SGB1814), *Streptococcus salivarius* (SGB8005) and *Klebsiella pneumoniae* (SGB10115). Shared strains are shown in black; persistent strains (found in the same infant at one and six months) are shown in gray. **E** Percentage of strains shared between mother-infant pairs stratified by infant stool collection time point. **F** Taxonomic composition of strains found to be persistent in the infant gut over time (up to six months). **G** Percentage of strains in infant stool at one month that persisted at six months divided by delivery mode. For each infant, persistence was calculated as the number of strains shared between the one- and six-month time-points, divided by the total number of strains detected at one month. Each dot is a mother-infant pair. The number of infants in each group is reported in brackets. *P*-value calculated with two-sided *t*-test, * for $p \leq 0.05$.

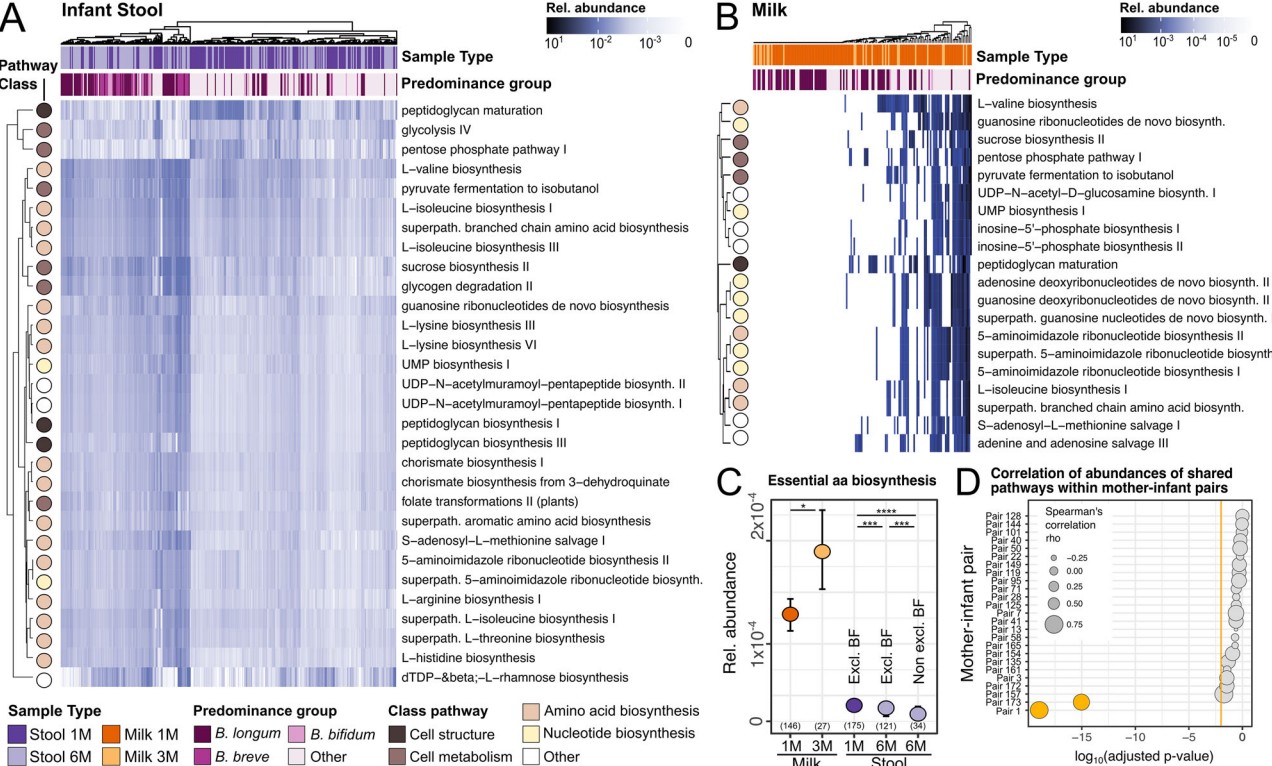

**Fig. 4 | Functional profiling of microbial metabolic pathways in breast milk and infant stool samples over time and across mother-infant pairs.** Most abundant pathways identified in **A** breastmilk at one and three months postpartum and **B** infant gut at one and six months of life. **C** Relative abundance of pathways involved in the biosynthesis of essential amino acids across sample type, collection time point and breastfeeding (BF). Error bars represent a 95% confidence interval calculated by bootstrapping (1000 times). *P*-values calculated using two-sided *t*-test, * for $p \leq 0.05$, ** $p \leq 0.01$, *** for $p \leq 0.001$ and **** for $p \leq 0.0001$. Reported *p*-values are adjusted using Bonferroni method. Bars represent the mean and sample

size is reported in brackets. **D** *P*-values of Spearman correlation between the abundance of all metabolic pathways shared between the maternal breast milk and infant stool samples for each mother-infant pair. *P*-values are calculated using a two-sided Spearman correlation test and corrected for multiple testing using Benjamini–Hochberg correction. Adjusted *p*-values are shown in $\log_{10}$ scale. Only the top 25 mother-infant pairs are shown. Circle size denotes the correlation coefficient (rho), and the orange line denotes the significance threshold (*p*-value = 0.01).

We then sought to investigate the relationship between the abundance of the metabolic pathways identified from microbes within each mother's milk with those identified in her infant's stool samples. We found a significant correlation between the metabolic pathways found in breast milk with those found in infant stool samples only for two of the mother-infant pairs for which we previously identified strain sharing events (BH-adjusted $p = 9 \times 10^{-20}$ and $p = 7.6 \times 10^{-16}$, for mother-infant pair 1 and 173, respectively, Spearman's correlation; Fig. 4D and Supplementary Fig. 12). When investigating specific pathways, rather than specific mother-infant pairs, we find that no pathway showed a

significant correlation in its abundance in milk compared to infant stool samples (Supplementary Fig. 13).

## The infant gut and the maternal milk microbiomes harbor a diverse landscape of antimicrobial resistance genes

The gut microbiome is a reservoir of antimicrobial resistance, yet its composition and longitudinal variability remain poorly characterized in infants[44,45]. Even less is known about the antimicrobial resistance genes (ARGs), collectively known as the resistome[46], present in human breast milk, and their transmission to the infant during lactation[21,47–49].

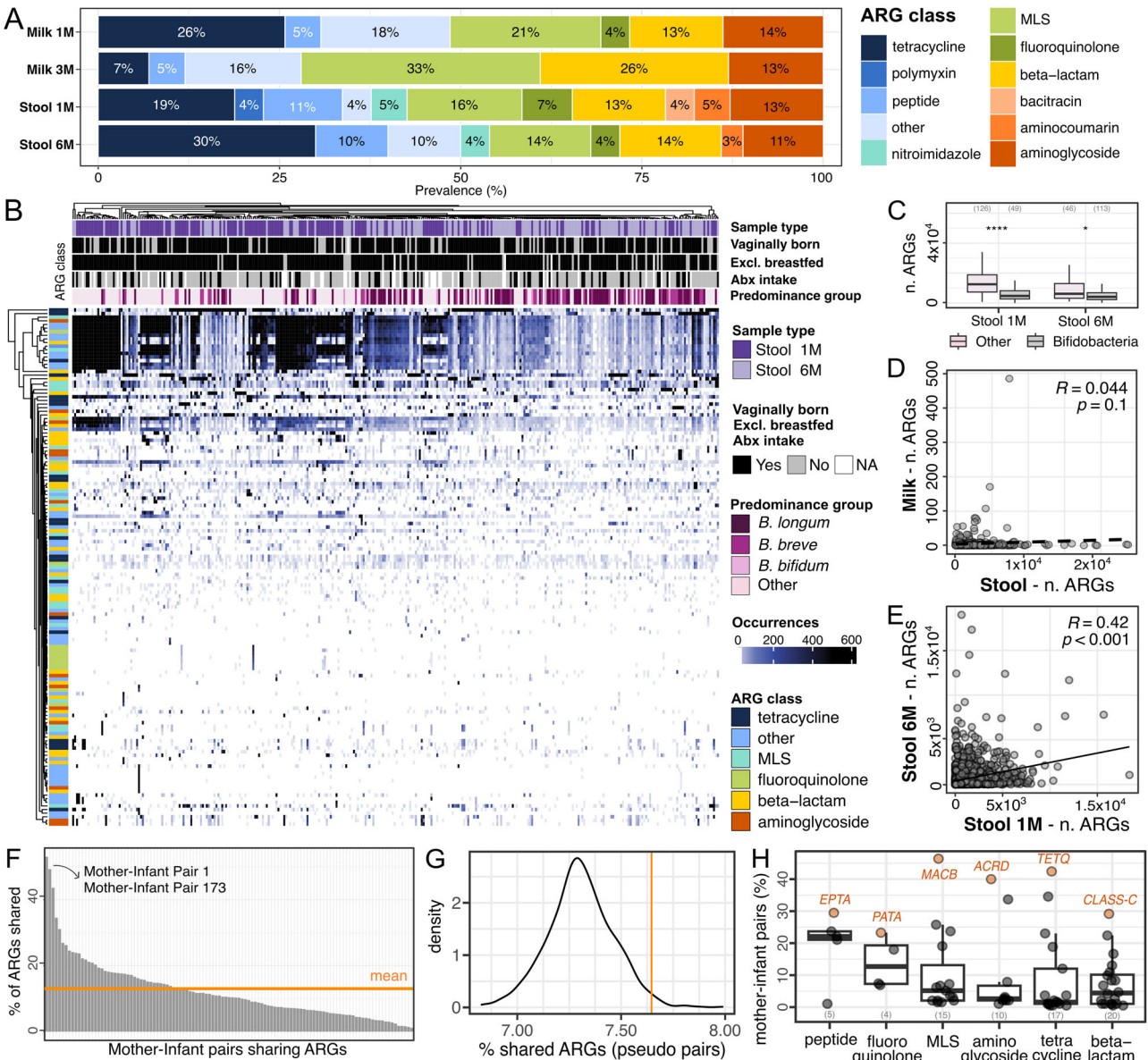

**Fig. 5 | Profiling and sharing of predicted antimicrobial resistance genes between maternal milk and the infant gut microbiome. A** Prevalence of the antimicrobial resistance genes (ARGs) classes predicted in maternal milk and the infant gut microbiome over time, calculated as percentage over the total number of detected ARGs for each sample type. **B** ARGs carriage in infant stool samples, divided by collection time point, infant metadata, predominance group and ARGs class. **C** Number of detected ARGs in infant stool samples dominated by bifido-bacteria and non-bifidobacteria species at one and six months of life. *P*-values calculated using two-sided *t*-test, ** for *p* ≤ 0.0001, * for *p* < 0.05. **D** Correlation between ARG carriage in maternal milk and infant stool samples, and **E** between infant stool samples at one month and six months. Each dot is a combination of a mother-infant pair and a predicted ARG class. Two-sided Spearman's R-values

(correlation coefficient) and *p*-values are reported. **F** Percentage of ARG genes shared between at least one maternal and one infant sample, for each mother-infant pair. Mean value is indicated by the orange line. The mother-infant pairs for which strain sharing events were identified are highlighted with the arrow. **G** Distribution of the percentage of shared ARGs between at least one maternal milk and one infant stool sample on permuted mother-infant pairs (pseudo-pairs). The mean value obtained from real mother-infant pairs is indicated by the orange line. **H** Percentage of mother-infant pairs sharing antimicrobial resistance genes, divided by ARG class. For each ARG class, percentage value was calculated on the total number of mother-infant pairs in which that ARG class was identified in at least one sample. The most frequently shared antimicrobial resistance genes per class are highlighted in orange. MLS Macrolides, lincosamides, streptogramines.

To investigate the resistome in the maternal milk and the infant gut, we used DeepARG[50], which leverages deep learning to predict anti-microbial resistance genes from metagenomic data. We then com-pared ARG classes across sample types and collection timepoints. Maternal milk and infant stool differed in terms of ARGs classes composition and prevalence (Fig. 5A, Supplementary Fig. 14A). In the milk, the most prevalent (27% on average) antimicrobial resistance class identified was against macrolide-lincosamide-streptogramin (MLS). Overall, maternal milk was characterized by a lower diversity

of detected ARG classes compared to infant stool samples ($p = 2.2 \times 10^{-16}$, *t*-test; Supplementary Fig. 14B, C). In both milk and infant stool samples, the diversity of ARGs classes increased over time, although this increase was statistically significant only in the milk ($p = 3.1 \times 10^{-4}$ and $p = 0.83$ for milk and stool samples respectively, paired *t*-test; Supplementary Fig. 14C).

The infant gut resistome was mostly dominated by resistance to tetracycline, MLS, aminoglycoside, and beta-lactams (Fig. 5A). Infants whose stool samples were dominated by bifidobacteria were

characterized by a significantly lower carriage rate of ARGs ($p = 7.6 \times 10^{-10}$ and $2.5 \times 10^{-2}$ at one and six months respectively, $t$-test; Fig. 5B, C). We found no significant difference in the resistome of infants that were born via C-section compared to those born via vaginal delivery ($p = 0.34$ and $p = 0.97$ at 1 month and 6 months, respectively, $t$-test), between those exposed to antibiotics and those that were antibiotics-naive ($p = 0.80$ and $p = 0.14$ at one month and six months, respectively, $t$-test), those that were introduced to solid foods at six months of age (0.71, $t$-test), nor between those exclusively breastfed at six months of age and those fed with a mixture of breast milk and formula ($p = 0.35$, $t$-test; Supplementary Fig. 14D–F). Nevertheless, we found extensive ARGs carriage in infants with no recorded pre-, during-, and postpartum exposure to antibiotics (Fig. 5B).

When comparing the overall resistome across all mothers and infants, we found no significant correlation between the ARGs found in milk and the ARGs found in the infant stool samples (R = −0.044, $p = 0.37$, Spearman's test; Fig. 5D). However, the infant gut resistome at one month was positively correlated with the gut resistome at six months of age (R = 0.42, $p < 0.001$, Spearman's test; Fig. 5E).

Considering sharing of ARGs between milk and stool samples within each mother-infant pair, we found that mother-infant pairs shared a significantly higher number of ARGs than what was expected by chance ($p < 0.016$, by permutation analysis, see Methods; Fig. 5F, G). Mother-infant pairs 1 and 173, for which strain sharing events were identified, were the pairs with the highest rate of shared ARGs between the mother's milk and her infant stool samples (Fig. 5F). On average, the most commonly shared antimicrobial resistance classes were associated with resistance to peptides, fluoroquinolone, and MLS, while the most commonly shared antimicrobial resistance genes were *MACB* (MLS class), *ACRD* (aminoglycoside class), and *TETQ* (tetracycline class) (Fig. 5H).

## Discussion

Although breast milk represents a critical source of nutrition for the developing infant, little is known about the milk microbiome, how it changes over time, its metabolic potential, and how it shapes the infant's gut microbiome. Answering these questions requires the use of high-resolution sequencing techniques, such as metagenomics. Here, we investigated the composition, strain sharing, functional potential, and antimicrobial resistance of the maternal breast milk and the infant gut microbiome in early life in predominantly exclusively breastfeeding mother-infant dyads. In line with previous studies, the milk microbiome yielded a lower number of reads and was characterized by a reduced microbial diversity compared to the infant gut microbiome[11,21,28]. The milk at one month postpartum, considered largely mature[18], was dominated by bifidobacteria, in particular *B. longum*, *B. breve*, and *B. bifidum*.

The bifidobacteria signature identified in our milk samples contrasts with some previous amplicon-based studies, which have typically reported the milk microbiome as being dominated by *Staphylococcus* and/or *Streptococcus*[29,51–54]. While bifidobacteria are widely recognized as infant gut commensals[55], their presence in human milk has also been documented in prior metagenomic studies[54,56,57] and confirmed through cultivation-based[57] and quantitative real-time PCR approaches[58,59]. This discrepancy between *Bifidobacterium*- and *Staphylococcus/Streptococcus*-dominated profiles may stem from amplification biases inherent to amplicon sequencing, as reported in other sample types[60,61], leading to the under-representation of bifidobacteria[62] and over-representation of *Staphylococcus*[63]. Notably, the use of universal PCR primers not specifically optimized for bifidobacteria can lead to their under-detection[62]. Furthermore, direct comparison of sequencing methods showed higher relative abundances of bifidobacteria in shotgun metagenomic data compared to amplicon-based data[54]. Additional contributing factors may include variation in DNA extraction protocols[64], particularly the inclusion and

extended duration (10 min) of the bead-beating lysis step, which has been shown to affect the detection and relative abundance of bifidobacteria[65]. In our study, DNA extractions for milk and infant stool samples were conducted independently by experienced personnel, therefore minimizing the likelihood of cross-contamination during this step. Beyond methodological considerations, the biological presence of bifidobacteria in breast milk is also supported by the ability of *B. longum* and other bifidobacteria to survive and grow at low-oxygen conditions[66–69]. In addition, oxygen-utilizing taxa commonly found in breast milk, such as *Streptococcus* and *Staphylococcus*, may reduce oxygen concentrations, facilitating the growth of bifidobacteria[33]. Finally, the presence in the milk of typical oral species, such as *S. salivarius*, is likely due to the possible transfer of microbes from the oral cavity of the infant to the breast milk during suckling, a process called retrograde flow[70].

While the breast milk and infant stool microbiomes were largely distinct in their microbial composition, they were both dominated by bifidobacteria. In particular, *B. longum* was the most prevalent and abundant species in both milk and infant stool samples, driving the limited overlap between these two sample types. While *B. longum*, *B. bifidum*, and *B. breve* were often coexisting and present at comparable abundances in milk, these species were largely mutually exclusive in the infant gut, suggesting a higher level of competition between bifidobacteria species in the infant gut compared to the milk. In line with previous findings[71], the gut microbiome of exclusively breastfed infants was characterized by higher prevalence and abundance of bifidobacteria compared to infants who ceased exclusive breastfeeding prior to six months of age, suggesting that prolonged breastfeeding supports the persistence and expansion of bifidobacteria in the infant gut microbiome. At the subspecies level, we found that both *BL. infantis* and *BL. longum* frequently co-existed in the same sample (in 81.4% of combined milk and infant stool samples), and that both were present at high prevalence (>75%) in infants at one month of age. Surprisingly, *BL. infantis* was already detectable at this early time point, in contrast to a recent study that found *BL. infantis* in infant stool samples only after 10 weeks of age[37]. Although that study's meta-analysis supported the delayed appearance of *BL. infantis* across multiple populations, samples collected from the United States cohort were limited to the first two weeks of life, preventing a direct comparison with our samples collected at one month of age. However, in line with the same study[37], we found that *BL. infantis* was significantly more abundant in infant stool samples at six months compared to one month and compared to *BL. longum* at six months of age.

At both the species- and strain-level, microbiome sharing between the mother's milk and the infant stool samples was more pronounced at one month postpartum than at six months, consistent with findings from previous studies on microbiome sharing between other maternal body sites and the infant gut[11,72]. This trend suggests that microbiome sharing between milk and the infant gut predominantly occurs during early infancy and may also reflect the effect of time, as a longer interval between sample collections could reduce the likelihood of detecting shared taxa. Overall, although there was a significant correlation between *Bifidobacterium* abundance in breast milk and infant stool in some mother-infant pairs, no statistically significant correlations emerged at the population level (e.g. across all mother-infant pairs). This discrepancy between individual dyads and population-level trends suggest that, while breast milk may serve as a reservoir for taxa like *BL. infantis*, successful colonization in the infant gut is likely influenced by additional factors (e.g., pre-existing microbial community composition, host genetics, or nutrient availability). Consequently, the relative abundance of bifidobacteria in breast milk does not necessarily predict their relative abundance in the infant gut, suggesting more complex mechanisms at play beyond the initial mother-to-infant microbial transmission. At the strain level, we found twelve cases of strain sharing between the mother's milk and her

infant's gut microbiome, in particular for the commensal species *B. longum*, *B. bifidum*, *V. parvula*, *P. vulgatus*, *S. salivarius*, *R. mucilaginosa* and for the pathobiont *K. pneumoniae*. Despite *K. pneumoniae* potential association with silent sepsis in infants[73,74] and with mastitis and milk loss in cows[75], the infants in our cohort showed no clinical manifestations nor had any of the mothers reported mastitis/breast inflammation at the time of their study visit and milk collection. Strain-level analysis also confirmed that the milk and the infant gut microbiome shared the same strains of species typically found in the oral cavity (*R. mucilaginosa* and *S. salivarius*). This could mean that either these species, initially present in the mother's milk, reach the infant gut by first colonizing the oral cavity, or alternatively it could be that the infant colonizes the breast milk shortly after birth with oral microbes acquired via other routes. As the first collection time point in this study was at one month postpartum, we were unable to discern between these two possible scenarios.

While strain sharing between the maternal and the infant gut has been extensively investigated[8,11], only one metagenomic study has so far, to the best of our knowledge, successfully identified strains shared between breast milk and the infant gut microbiome[57]. Our results provide evidence that strain sharing between the infant gut and the maternal breast milk occurs, even if at a rate considerably lower than what was previously found between the infant and the maternal gut[11]. The fact that we were able to reconstruct strains from only ten taxa in our milk samples, and yet all were found to be shared between mother's milk and infant gut, indicates that our estimate for strain sharing between breast milk and infant gut is likely an underestimate due to the low number of microbial reads in the milk samples. This is a common challenge with this type of sample[28]. While prebiotics (e.g. HMOs) have been widely studied for their potential role in shaping the infant gut microbiome[37,76–78], our findings support the transmission of milk microbial strains as a contributing factor to infant gut microbiome assembly.

Strain-level analysis also revealed rates of microbial persistence in the infant gut comparable with what previously reported[79], with 19% of the strains found in the infant stool samples at one month persisting to six months of age, and 25% of strains shared between the mother's milk and the infant gut at one month being identified also at six months of age. We also found that delivery mode impacted microbial persistence rate in the infant gut: vaginally born infants were characterized by significantly higher strain persistence over time than those born via C-section. This result supports previous amplicon and species-level metagenomic studies that associated C-section with a more diverse and less stable infant gut microbiome[15,80–82].

Resistome analysis showed that both the infant gut and the maternal milk harbor a diverse landscape of antimicrobial resistance gene classes, mostly dominated by resistance genes for tetracycline, aminoglycoside, and macrolide-lincosamide-streptogramin, even in infants that had no recorded exposure to pre-, intra-, and postpartum antibiotics. Tetracycline is not prescribed in pregnancy and its use is not recommended in children younger than 8 years due to potential permanent discoloration of teeth[83]. Still, tetracycline resistance was the most abundant antibiotic resistance class in both milk and infant stool samples at one month. Aminoglycoside (e.g. gentamicin and streptomycin) and macrolides (e.g. azithromycin and erythromycin) are widely used in the perinatal period[49], therefore resistance is potentially acquired via antibiotic exposure. Milk and infant stool samples at one month were characterized by higher prevalence and diversity in resistance classes, compared to later time points. In addition, the majority of ARGs found in the infant gut at one month were associated with predominance of non-bifidobacteria species, in particular *E. coli*. The presence of antibiotic resistance in the newborn gut has been reported in previous studies[44,45,84], with indications of resistance transmission from the maternal gut to the infant gut via mobile genetic elements[85]. Our results show a significant overlap between the resistome of infants and that of their mother's milk, and that this overlap was particularly pronounced when evidence of strain sharing was found. This suggests that the infant gut resistome is likely influenced to some degree by acquisition from breastmilk, in addition to other mechanisms, such as mobile genetic elements, vertical strain transmission and environmental exposure, and from other maternal body sites, such as gut[85].

This study has several limitations. Milk sampling was performed via the use of breast pumps, which could potentially impact milk microbiome composition[86]. The low microbial load in breast milk samples limited our ability to detect strain sharing between milk and the infant gut, particularly in the milk samples collected at three months, suggesting that the observed sharing patterns likely represent an underestimate. Notably, the samples with the higher number of strain sharing events corresponded to mother-infant pairs 1 and 173, which ranked first and third, respectively, in the number of reads mapped to MetaPhlAn4 marker gene database. These findings suggest that higher microbial yield and deeper sequencing depth for milk samples should be a priority for future metagenomic studies. The definition of a standardized and optimized protocol for human breast milk extraction to maximize microbial yield should also be prioritized by the field. In addition, as we sampled only milk and infant stool samples, we could not confirm that the strain sharing events we identified were indeed cases of microbial transmission from the maternal milk to the infant gut, rather than cases of strain acquisition by both the mother and the infant from external sources not investigated in this study. Finally, although infants were born across multiple hospitals, which could potentially influence microbial composition, all samples were collected using the same procedures, and the taxonomic composition of the samples did not cluster based on the sampling location.

In this work, we characterized the microbiome composition, function and antimicrobial resistance potential of the breast milk of mothers and the gut microbiome of their infants during the first six months postpartum. We found evidence of strain- and antimicrobial resistance gene-sharing between mother-infant pairs. Taken together, our results indicate that the maternal breast milk plays a role in infant gut microbiome and resistome establishment, development and temporal stability. Our results represent an important step towards the strain-level characterization of the maternal milk microbiome in relation to the infant's gut and its better representation in public repositories.

## Methods
### Sample collection
Participants in this study were enrolled as part of the Mother and Infants Linked for health (MILk) cohort[87–91]. All mothers were enrolled prenatally at the University of Minnesota in collaboration with HealthPartners Institute (Minneapolis, MN), and provided written informed consent. The study was approved by the Institutional Review Boards of the University of Minnesota, the University of Oklahoma, and HealthPartners Institute (STUDY00009021), and conducted in accordance with all relevant guidelines and regulations. All participant data included in this study were anonymized. Sample identifiers were randomized, infant age was recorded at the time of the study visit, and no hospital, location, or other personally identifiable information is included in the supplementary materials. Eligibility criteria included a healthy, uncomplicated pregnancy, and intention to exclusively breastfeed. All mothers were aged between 21 and 45 years, were not diabetic and non-smokers, and delivered a full-term infant. No case of breast infection or mastitis was reported during the milk sample collection. All infants were singletons and born at term with a birth weight that was appropriate for their gestational age, and were exclusively breastfed to at least one month of age (Supplementary Data 1,2 and Supplementary Fig. 1). All relevant metadata were collected via the

hospital's electronic medical health records and via questionnaires during sample collection (Supplementary Data 1, 2 and Supplementary Fig. 1). Ethnicity and race of the participants are included in Supplementary Data 2. Samples were collected between 2014 and 2023. Breast milk samples were collected at one and three months postpartum. Milk collection was performed as follows: mothers fed their infant from one or both of their breasts, until the infant was satisfied. After two hours, the milk was collected from the right breast using a hospital grade electric breast pump (Medela Symphony; Medela, Inc., Zug, Switzerland) using sterilized tubing and in a sterilized milk collection bottle, until cessation of production. Each milk sample was gently mixed, and its volume and weight were recorded. Aliquots were stored at −80 °C within 20 min of collection and kept at that temperature until RNA/DNA extraction. Infant stool samples were collected at one and six months of age. Stool samples were either collected from diapers during a study visit or at home by the mother using stool collection kits designed for microbial DNA extraction provided by the Gale laboratory. In case of collection during a study visit, the sample was immediately frozen at −80 °C, while in case of home collection the sample was stored in 2 ml cryovials with 600 μl RNALater (Ambion/Invitrogen, Carlsbad, CA), and later stored at −80 °C upon arrival to the lab at the University of Minnesota, as described also in refs. [87–91].

### DNA extraction and metagenomic sequencing

Starting volume for extraction was 1 ml for breast milk, and the weight comparable to a pea-sized scoop for the infant feces. DNA extraction was performed with PowerSoil Pro kit (QIAGEN, Germantown, MD). Cell lysis was carried out by bead-beating for 10 min, followed by elution with 100 μl of the provided elution solution. The extracted DNA was then stored in microfuge tubes at −80 °C and subsequently used to construct libraries for metagenomic shotgun sequencing using the Illumina Nextera XT 1⁄4 kit (Illumina, San Diego, CA, United States). DNA extractions for milk and infant stool samples were conducted independently by experienced personnel. Metagenomic shotgun sequencing libraries for milk and infant stool samples were then sequenced concurrently on an Illumina NovaSeq system (Illumina, San Diego, CA) by the University of Minnesota Genomics Center. A small subset (n = 30) of milk samples at 1 month postpartum was selected to be re-sequenced at the target depth of 30 M reads per sample to improve detection of species, strains, functional pathways and ARGs shared between mothers and their infants. For those 30 samples, raw sequences from the two batches were concatenated before pre-processing and analyzed together with the rest of the samples in the downstream pipeline described below.

### Quality filtering and removal of human reads

Host DNA was removed using paired-end mapping with Bowtie2[92] version 2.2.4 against a human reference genome hg38. Unmapped paired-end reads were filtered using SAMtools[93] version 1.9 with following parameters "samtools view -bS, samtools view -b -f 12 -F 256, samtools sort -n -m 5G -@ 2, samtools fastq -@ 8 −0 /dev/null -s /dev/null -n". BEDtools[94] was then used to convert the bam files to fastq files containing the non-human paired-end reads. Then, adapter sequences were removed and the samples were filtered and trimmed using the default parameters of Trimmomatic[95]. Host DNA was removed using the most complete version of the human genome as reference (T2T-CHM13, including Y chromosome, available here: https://s3-us-west-2.amazonaws.com/human-pangenomics/T2T/CHM13/assemblies/analysis_set/chm13v2.0.fa.gz). Mean host DNA content in breast milk samples was 97.3%. FastQC[96] was used to analyze the quality of the metagenomic reads. Mean PHRED of the score was 33. Samples that had less than 1000 reads that mapped to the Metaphlan4[40] database (mpa_vJan21_CHOCOPhlAnSGB_202103) were discarded. Further pre-processing was conducted using the fastp[97,98], where reads underwent

length filtering at a minimum of 75 bp, quality trimming, additional adapter trimming, and poly-G/poly-X trimming with the parameters "fastp -i Input R1 -o Output R1 -I Input R2 -O Output R2 --detect_adapter_for_pe -r -l 75 -x". After pre-processing, milk samples yielded 2.5 ± 1.36 million reads per sample, while stool samples yielded 21.4 ± 8.01 million reads per sample.

### Species-level taxonomic profiling and *B. longum* subspecies detection

Species-level taxonomic profiling was based on marker genes using MetaPhlAn4[40], with the following parameters: "--bt2_ps sensitive" text. MetaPhlAn4 infers taxonomic prevalence and abundance by using unique marker genes for 26,970 species-level genome bins[40]. As MetaPhlAn4 has no marker genes to identify and distinguish *B. longum* subspecies (mainly *BL. infantis* and *BL. longum*), we used a modified version of the standard MetaPhlAn4 database. Details of the modified database can be found in the original publication[37], while the database can be found on GitHub (https://github.com/yassourlab/MetaPhlAn-B.infantis/). Unclassified *B. longum* subspecies was not included in the subspecies analyses. Merged abundance table was created using the dedicated MetaPhlAn4 utils script. Contaminants found in the blank samples as well as additional known contaminant species were removed before downstream analysis. Milk samples with no identifiable bacterial taxa via MetaPhlAn4 were excluded from downstream analysis (n = 26). MetaPhlAn4 profiles are available in Supplementary Data 3. Heatmaps for species-level taxonomic composition were generated with ComplexHeatmaps[99,100], using euclidean distance for hierarchical clustering. Correlation results shown in Supplementary Fig. 10 were calculated using the sm_statCorr() function part of smplot2, using Pearson as correlation method.

### Maternal secretor status

Mothers with an inactive fucosyltransferase 2 (*FUT2*) gene are classified as non secretors. Secretor status for the mothers in this study was obtained from[91] and was calculated based on the concentration of 2′-fucosyllactose in breast milk.

### Strain-level profiling and strain sharing

Strain-level profiling was performed using StrainPhlAn4[40], based on single-nucleotide variant calling, using default parameters. Strain-sharing events between samples were identified based on the output of the "strain_transmission.py" script, which is part of the StrainPhlAn4 release. Detailed step-by-step description is available in https://github.com/biobakery/MetaPhlAn/wiki/Strain-Sharing-Inference. Briefly, the "strain_transmission.py" script takes as input the phylogenetic tree computed by StrainPhlAn4 for each species-level genome bin (SGB), SGB ID, and metadata table containing information on subjectID, relation (e.g. mother-infant pair), and sampling time point. Pre-computed thresholds provided in "VallesColomerM_2022_Nov19_thresholds.tsv" were used when possible, otherwise the default threshold of 0.03 was provided to the script. The script outputs a list of all sample pairs in which strain sharing was detected. The collated output of the "strain_transmission.py" script is available as Supplementary Data 4. Phylogenetic trees were visualized with iTOL[101].

### Alpha and beta diversity

Both alpha diversity and ordination plots were computed using the "diversity" and "vegdist" functions, respectively, included in the "vegan" R package (v2.6-4).

### Functional profiling

Functional prediction was performed with HUMAnN3[43]. Biosynthetic potential of essential amino acids were calculated by searching the output pathways for the following keywords: "L-histidine biosynthesis", "L-isoleucine biosynthesis", "L-isoleucine biosynthesis", "L-lysine

biosynthesis", "L-methionine biosynthesis", "L-phenylalanine biosynthesis", "L-threonine biosynthesis", and "L-valine biosynthesis". Profiles relative abundance plots show 95% confidence interval using bootstrapping with 1000 repetitions using the Hmisc R library (v5.0-1). Raw functional profiles are available in Supplementary Data 5.

## Antimicrobial resistance genes prediction

Antimicrobial resistance genes were predicted from raw metagenomic short reads using DeepARG v1.0.2[50]. DeepARG leverages deep learning models to identify over 30 antimicrobial resistance classes. Predicted resistance is classified as "predicted" or "potential" by the models. To reduce false positives detection, ARG predictions classified as "multi-drug" or "unclassified" were excluded from downstream analyses and a minimum 95% identity threshold was applied. Raw DeepARG profiles are available in Supplementary Data 6. Correlation results shown in Fig. 5D, E were calculated using the sm_statCorr() function part of smplot2, using Spearman as correlation method. Permutation analysis in Fig. 5G was performed as following: maternal milk sample names were randomly permuted, while data and the infant sample names were preserved, generating pseudo-mother-infant pairs, and the mean number of antimicrobial resistance genes shared between the pseudo-mother-infant pairs was calculated. This was repeated 1000 times, and the distribution of the mean values obtained from pseudo-mother-infant pairs was compared to the mean value obtained from the real mother-infant pairs.

## Statistics & reproducibility

Statistical analysis was done in R[102] version 4.2.2 (2022-10-31). All figures if not indicated otherwise were drawn with ggplot[103] version 3.4.1. All analyses have been performed with open source software referenced in the Methods section. Statistical significance was set at $p < 0.05$, and Benjamini-Hochberg false discovery rate (FDR) correction was applied where appropriate to account for multiple testing. Unless otherwise stated, boxplots show the median as the center line, the interquartile range (25th to 75th percentile) as the box, whiskers extending to 1.5× the interquartile range, and outliers are not shown. No statistical method was used to predetermine sample size and no data randomization was performed. No data were excluded from the analyses. This was an observational study and participants received no interventions. Participants were not assigned to experimental groups; instead, they were grouped retrospectively based on metadata (e.g., breastfeeding status). The investigators were not blinded to allocation during experiments and outcome assessment.

## Reporting summary

Further information on research design is available in the Nature Portfolio Reporting Summary linked to this article.

## Data availability

The raw metagenomic data generated in this study have been deposited in the NCBI Sequence Read Archive (SRA) database under BioProject accession numbers PRJNA1019702 and PRJNA1198101. The metadata associated with this study are provided in the Supplementary Information file. All supplementary tables are also available on Zenodo [https://zenodo.org/records/17317218].

## Code availability

Code is available on github https://github.com/blekhmanlab/milk_infant_microbiome.

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

## Acknowledgements

We thank the MILk Study staff and participants (https://www.milkresearch.umn.edu), and members of the Albert and Blekhman labs for support and helpful discussions related to this project. This work could not have been completed without the scientific collaboration of the Health Partners Institute (Bloomington, MN) and the University of Minnesota Medical School, Department of Obstetrics, Gynecology, and Women's Health. This work was supported by the resources and staff at the Masonic Institute for the Developing Brain at the University of Minnesota (https://www.midb.umn.edu), the University of Oklahoma Health Sciences Center Harold Hamm Diabetes Center (https://www.ouhealth.com/find-a-location/ou-health-harold-hamm-diabetes-center) and the University of Minnesota Genomics Center (https://genomics.umn.edu). We are thankful for the computational resources provided by the Minnesota Supercomputing Institute (https://www.msi.umn.edu/) and by the University of Chicago Research Computing Center (https://rcc.uchicago.edu/). The MILk Study was supported by NIH/NICHD grant R01HD080444 (E.W.D. and D.A.F.). Additional support for this study was provided by the Masonic Cross Departmental Research Grant from the University of Minnesota Department of Pediatrics (F.W.A., R.B., E.W.D., and C.A.G.); the Masonic Children's Hospital Research Fund Award (C.A.G., E.W.D., and D.K.); NIH/NICHD grants R01HD109830 (R.B., F.W.A., E.W.D., C.A.G., and K.E.J.), R21HD099473 (C.A.G. and E.W.D.), F32HD105364 and K99HD113834 (K.E.J.); and a Faculty Research Development Grant from the University of Minnesota Office of Academic and Clinical Affairs (C.A.G., E.W.D., and D.K.).

## Author contributions

Conceptualization, R.B., C.A.G., E.W.D., and D.K.; Sample and metadata collection, E.W.D, S.G., and C.A.G; Samples processing and data analysis, P.F., M.A. and M.R.; Writing – original draft, P.F.; Writing – review & editing, P.F., M.A., K.J., M.R., T.H., D.A.F., F.W.A., C.A.G., E.W.D., and R.B.; Supervision, P.F. and R.B.; Funding acquisition, E.W.D., D.A.F., and R.B. All authors read and approved the final manuscript. The Authors declare no conflict of interest.

## Competing interests

The authors declare no competing interests.
