## [Transparent Peer Review file · Nature Communications]

Assembly of the infant gut microbiome and resistome are linked to bacterial strains in mother's milk

Corresponding Author: Professor Ran Blekman

Version 0:

Reviewer comments:

Reviewer #1

(Remarks to the Author)

The current study by Allert & Ferretti and colleagues attempted to link bacterial strains in mother's milk to the infant gut microbiome. The figures are beautifully presented and the manuscript is well written with good flow. I have some important comments below for the authors consideration.

The title states that the infant gut microbiome is linked to bacterial strains in mother's milk, but the authors only identified two strains in milk. Similarly, the title suggests infant gut microbiome is linked to bacterial function in mother's milk, but the authors state "we find that no pathway showed a significant correlation in its abundance in milk compared to infant stool samples". Thus, the title as it stands is misleading.

Indeed, the main novelty of this paper over previous milk microbiome studies was the application of metagenomic sequencing, but from the methods it reads as though the metagenomic approach of milk largely failed - the threshold for including a sample was very low (>500 reads mapping to a bacterial marker gene database)?

How much stool weight and milk volume were used in the DNA extractions? This is especially important for low-biomass milk samples. It also appears as though the extraction method was not optimized for low-biomass samples, e.g., the elution volume of 100ul will greatly reduce DNA concentration. This would likely be permissible for amplicon-based sequencing, but is not optimal for metagenomics.

Given the potential issues with the methods, it is important to note the current results are in disagreement with all other published milk microbiome studies. There are several published milk microbiome studies now that all show dominance of Bacillota (Firmicutes) phylum, specifically Staphylococcus and/or Streptococcus. In contrast, the current study shows the dominance of Bifidobacterium species in milk (and infant stool), which is considered a gut commensal. Considering this study sequenced both milk and stool, the latter being expected to have high Bifidobacterium, I have concerns there may have been sample cross-contamination e.g., during extractions, library prep, or sequencing bleedover. This needs to be checked, for instance, the authors could re-extract a random selection of milk samples and perform metagenomics and 16S rRNA gene sequencing. Metagenomics will confirm the results of the current study and 16S rRNA gene sequencing will demonstrate if the lack of concordance with previous literature is due to the sequencing method (as most previous milk microbiome studies used amplicon-based sequencing).

That the authors could not link the transmission of Bifidobacterium strains from milk to the infant is a weakness of the current study. Given the shortcomings of the metagenomic approach, could they culture Bifidobacterium from both sample types and confirm this with WGS?

Related to the above, the high abundance of a strict anaerobe like Bifidobacterium has not been reported in milk and is unexpected since milk is not anaerobic. Thus, proving the ability to culture Bifidobacterium from milk would alleviate some concerns with the unusual finding that Bifidobacterium species dominate human milk.

Mother-infant pairs 1 and 173 were the only pairs found to have strain sharing events, and subsequent analysis found these pairs were most related in shared function and ARGs. Were these milk samples among the deepest sequenced / had the highest number of reads mapping to the marker gene database?

The metadata is not very comprehensive or granular. For example, much of the analysis focuses on breastfeeding at month 1 and 6, defined as exclusive or mixed. However, we know from previous studies the amount of human milk in the diet, whether the milk was expressed or not, what % of the overall diet was human vs bovine milk, etc impact infant gut microbiome and Bifidobacterium. Without this extra granularity in the metadata, the current results of infant gut microbiome largely support other similar studies, and the current study has a smaller sample size than many recent publications.

Other specific comments

- Introduction lines 63+: Another reason the milk microbiome is understudied compared to e.g., the gut, is because using such samples for research (and therefore reducing the volume received by the infant) has important implications which are not relevant when studying e.g., a stool sample.
- It needs to be clear in the first lines of the results where the mothers/infants were recruited from geographically.
- The n needs to be clearer in the figure legends. E.g., Figure 2 since panel B will include only subjects with both samples, the n will not be all mothers, whereas in panel C I suspect all mothers are included. The exact n in each analysis should be in the figure and/or legend.
- Line 119: use of "strains" is not appropriate. Suggest species. Lines 141-162: The majority of this section is describing Fig 2B with words, but the figure clearly shows the %s and is easy to follow, so the text does not add anything. It also distracts from the actual analysis that follows.
- Line 158: "particularly pronounced reduction" – was this significant?
- Line 163-176: it could be clearer this is stool microbiome
- Extended data 1 is difficult to follow and would be better summarised as a table.
- Considering the large % of host reads in the milk dataset, could the authors use this to determine secretor status of mother?
- Line 334: Bifidobacterium are not oral bacteria. This is related to my earlier comment that skin/oral facultative such as Staphylococcus and/or Streptococcus are expected in human milk, not gut bacteria like Bifidobacterium

(Remarks on code availability)

Reviewer #2

(Remarks to the Author)

In this study, Allert et al. dive into the connection between the breast milk microbiome and the infant gut microbiome. The authors show that Bifidobacterium, and specifically *B. longum*, overlaps between milk and infant stool samples. Next, they find two strain sharing events between an infant's gut and its mothers breast milk, and show that these dyads also have a high percentage of ARG sharing. The study overall suggests a connection between breast milk microbiome and the influence it has on the infant gut microbiome.

Major comments:

- The introduction is somewhat lacking, specifically regarding previous literature on the breast milk microbiome and on Bifidobacterium species:
 - a. The authors mention that most previous research on the breast milk microbiome used 16S-rRNA sequencing, however what is found in the current literature should be elaborated on. For example, does previous work show a high abundance of Bifidobacteria in breast milk? Do the results that are found in this study match what is already known?
 - b. Bifidobacterium is a common genus found in the infant gut and is known to utilize human milk oligosaccharides (HMOs) along with many additional health benefits to the infant. However, no focus is given in the introduction to Bifidobacterium and specifically to *B. longum* which is one of the focuses of this study.

-One of the main messages of the paper is that there is a compositional overlap between breast milk and the infant gut, specifically of Bifidobacteria and *B. longum*.

- a. Overlap between breast milk and infant stool should be done at the mother-infant pair level as well as the community level.

The overlap shown is only at the community level and not for individual dyads, therefore it is hard to understand if Bifidobacteria species are just abundant in both types of samples or perhaps this might indicate sharing of species between mother and infant. Overlap of Bifidobacteria at the dyad level will allow further insight. For example, in Fig. 2A it is shown that *B. longum* is the predominant and abundant species in infant gut and breast milk samples, however is this true for matched samples from the same dyad? Do infants with abundant *B. longum* in their gut have mothers with abundant *B. longum* in the breast milk?

- b. Sharing analysis should be added for the species level, rather than just the strain level. The authors elaborate and find two events of strain sharing between breast milk and the infant gut. It is clear that on the strain level it is hard to find sharing events due to the low coverage in the milk samples, however there is no reference to sharing at the species level. Perhaps analyzing shared species and specifically Bifidobacterium species between breastmilk and the infant gut will allow further insight into the possible sharing events.

- c. It is not clear how strain sharing events were defined. For example in Fig. 3A for mother-infant pair 1, how was it determined that the three strains shown are the same strain? What is the distance between the strains?

- This study focuses on *B. longum*, a common infant gut commensal, however it is clearly known that there are two main different subspecies within this clade that are found in infants: Bifidobacteria longum subsp. infantis and Bifidobacteria longum subsp. longum. There are clear differences between these subspecies and their ability to function in the infant gut,

specifically in the context of HMOs. Therefore I suggest using a method that will allow differentiation between these subspecies, and focus on the subspecies that overlaps between breast milk and the infant gut microbiome. This can be done using the existing metagenomic data (Ennis et al. 2024, Casaburi et al. 2021), or qPCR with subspecies-specific primers. It will also be interesting to see if *B. infantis* is found in breast milk samples.

- Throughout the manuscript and in the title it is mentioned how the stability and dynamics in the infant gut microbiome is linked to the bacteria in breast milk, however this statement should be changed as currently, it is not proven in the paper.
- a. When discussing stability (Fig. 2) the analysis is performed separately for breast milk and the infant gut. The main focus is on the infant gut microbiome with differentiation between infants that were exclusively breastfed at 6 months and those that weren't. How does the milk microbiome influence the stability of the infant gut? Is it only being breastfed or not?
- b. Are there any direct effects of breast milk microbial composition/ specific bacteria that influence the stability or dynamics in the infant gut microbiome? Was this checked and just not significant?

Minor comments:

- Fig 1E- Axis titles should be Pco1 and Pco2 (instead of PC1 and PC2).
- Fig 2D- is this figure only for samples that have the specific bacteria? If yes, this should be mentioned in the caption. All samples had *B. longum* at 6 months?
- In line 164 "We found that *B. longum* and *E. coli* were more abundant at six months compared to one month", however in Fig. 2E *E. coli* seems to decrease over time. The same is true for exclusively breastfed infants according to Fig. 2F (line 169).
- Fig 3C- it would be important to mention what species have shared strains between unrelated infants (or what are the main species with shared strains).
- Fig. 4D- On one hand it is mentioned that strain sharing events are probably missed due to low coverage in breast milk samples, on the other hand significant correlation between metabolic pathways in breast milk and stool samples were found only in dyads where shared strains were found. Can this lack of significant correlation in most dyads be explained by the low coverage in breast milk samples? Perhaps the correlation in pairs 1 and 173 is due to the specific bacteria shared within the pair (*B. longum* and *K. pneumoniae* respectively)- do these bacteria have a high relative abundance in the relevant samples? In addition, do pairs 1 and 173 have a higher coverage in breast milk samples which can explain this?

(Remarks on code availability)

Version 1:

Reviewer comments:

Reviewer #1

(Remarks to the Author)

The additional work substantially improves the quality and robustness of the work.

(Remarks on code availability)

Reviewer #2

(Remarks to the Author)

The authors have addressed most of the main comments. Milk samples were sequenced more deeply, separated between *B. longum* and *B. infantis*, and added relevant points and limitations regarding their study to both the Introduction and Discussion. Figure 3 was refined substantially, including species-level sharing, and deeper sequencing allowed them to identify more strain-sharing results and define strain-sharing events more clearly.

However, since the milk microbiome is of such low biomass, the authors should demonstrate how the samples that were sequenced more deeply match and correlate with their previous results from these same samples, as this would strengthen confidence in the metagenomic analysis of these low-biomass samples.

(Remarks on code availability)

Summary for the Reviewers and the Editor

We are extremely thankful to the Reviewers for their useful comments and suggestions. For the purpose of this revision, we have generated new data by generating new deep metagenomic sequencing data for 30 of the breast milk samples, which overall significantly strengthened our previous results. We have addressed all points raised by the Reviewers, and we have included several new analyses. We provided a point-by-point response below. In summary:

1. To address the limited microbial readout from milk samples in our initial analysis—an issue that may have hindered the detection of strain-sharing events with the infant gut—we generated new, deep metagenomic sequencing data for a subset of the milk samples. Specifically, we re-sequenced 30 milk samples collected at one month postpartum (available on ENA under accession number PRJNA1198101), targeting a sequencing depth approximately six times greater than in the original dataset. This additional sequencing effort enabled the detection of a significantly greater number of strain-sharing events between mothers and their infants increasing, from $n=2$ in the initial submission to $n=12$ in this updated version. As a result, we were able to strengthen and expand the section on mother-infant strain sharing (see **new Figure 3D and 3E**).
2. We performed a new MetaPhlAn4 taxonomic profiling leveraging a recently published database¹ that differentiates between different *B. longum* subspecies, as these subspecies can differ significantly in their gene content and in their ability to utilize human milk oligosaccharides. This new analysis allowed us to identify novel trends in both milk and infant stool samples that were not detected in the initial analysis of this data. For instance, we found that *BL. infantis* and *BL. longum* differed in their prevalence and relative abundance in both breast milk and infant stool samples, as well as in their persistence in the infant gut from one to six months of age. These new results are now shown in the **new Figure 3B-C and 3F**, and in the **new Extended Data 6, 7, 9A-B and 10**.
3. While re-analysing our results on strain-sharing events between unrelated infants, we identified and corrected an issue with our code. Upon fixing the identified issue, we could not confirm our initial results on strain sharing between unrelated infants. We have therefore removed Figure 3C and its associated text from the updated version of the manuscript.
4. We included a novel analysis to investigate the persistence of microbial strains in the infant gut over time. This analysis identified which strains were most consistently retained during the first six months of life and revealed that vaginally delivered infants harbored a higher number of persistent strains compared to those born via C-section. These findings are presented in the **new Figure 3F and 3G**.
5. We have also included, as suggested by the Reviewers, new analyses on the microbial species shared between mothers and their infants, and on the correlation of bifidobacteria abundance in mother-infant pairs. For instance, we found that on average 10.1% of taxa

found in infant stool samples at one month of age were also found in their respective mother's milk. These results are shown in the **new Figure 3A, 3B and 3C**. These species-level results complement the strain-level analysis shown in the updated Figure 3D and in the **new Figure 3E**, providing additional insight on the microbiome overlap between mothers and their infants. Furthermore, we included additional metadata relative to the HMOs secretor status of the mothers included in this study. We found that maternal secretor status did not impact microbial diversity or composition of the milk microbiome, nor with differences in strain persistence in infant stool samples. These results are shown in the **new Extended Data 2**.

In summary, we have added the following new figures: 3A–C, 3E–G, and Extended Data Figures 2A–B, 6A–C, 7A–F, 9A, and 10. We have also updated the following figures: Figures 1B-C, 1E-F, 2A-G, 3D, 4B-D, 5A, 5D, 5F-H, Extended Data Figures 3, 4A-B, 5, 6A-C, 8, 11B, 12, 13 and 14A-C.

Reviewer #1, comments to the authors:

Major Comments:

The current study by Allert & Ferretti and colleagues attempted to link bacterial strains in mother's milk to the infant gut microbiome. The figures are beautifully presented and the manuscript is well written with good flow. I have some important comments below for the authors consideration

We thank the Reviewer for their time and their valuable feedback in improving this manuscript.

The title states that the infant gut microbiome is linked to bacterial strains in mother's milk, but the authors only identified two strains in milk. Similarly, the title suggests infant gut microbiome is linked to bacterial function in mother's milk, but the authors state "we find that no pathway showed a significant correlation in its abundance in milk compared to infant stool samples". Thus, the title as it stands is misleading.

Response 1.1. We agree with this comment, and thank the reviewer for pointing this out. To address this, we have changed the title of the paper to "*Assembly of the infant gut microbiome and resistome are linked to bacterial strains in mother's milk*", which we believe better reflects the paper. In addition, we note that we have generated new, deeper sequencing data from some of the same samples; see **Responses 2.4, 2.6, 2.7, 2.8, and 2.9** below for a full description of these new data and results. Importantly, the newly generated deep metagenomic data provided a more comprehensive analysis of the microbial composition of breast milk, enabling a more robust identification of strains. This, in turn, strengthened our findings regarding strain sharing between mother-infant pairs. Consequently, we decided to retain this aspect in the title. Additionally, the updated title now reflects the inclusion of resistome results. Specifically, we found a significant correlation between the resistome profiles of infants and their mothers within specific mother-infant pairs, as shown in the updated version of **Figure 5F** (see below).

Figure 5. (F) Percentage of ARG genes shared between at least one maternal and one infant sample, for each mother-infant pair. Mean value is indicated by the orange line. The mother-infant pairs for which strain sharing events were identified are highlighted with the arrow.

Indeed, the main novelty of this paper over previous milk microbiome studies was the application of metagenomic sequencing, but from the methods it reads as though the metagenomic approach of milk largely failed - the threshold for including a sample was very low (>500 reads mapping to a bacterial marker gene database)? How much stool weight and milk volume were used in the DNA extractions? This is especially important for low-biomass milk samples. It also appears as though the extraction method was not optimized for low-biomass samples, e.g., the elution volume of 100ul will greatly reduce DNA concentration. This would likely be permissible for amplicon-based sequencing, but is not optimal for metagenomics.

Response 1.2. We thank the Reviewer for bringing up this important point. We agree that human breast milk represents a challenging sample to work with, due to its low microbial biomass. The extraction kit used in this study was carefully chosen to maximise the number of microbial reads obtained from milk samples. PowerSoil Pro, the kit we used, was recently confirmed to be the best choice for microbiome analysis from milk samples by two independent publications^{1,2}. We agree that the discrepancy between the microbial reads recovered from milk and stool samples might have reduced our ability to identify strain sharing events between breast milk and the infant gut.

To address this issue and improve the microbial readout from breast milk samples, we have generated new, deep metagenomic shotgun sequencing for 30 of the milk samples at a depth of 30 million reads per sample, which is 6 times deeper than the data used in the previous version of our paper. This additional sequencing data yielded a substantially higher number of microbial reads, which allowed us to perform more robust species- and strain-level analyses and overall strengthened our results on milk microbiome composition, antimicrobial resistance carriage in milk samples, and milk-infant gut strain sharing (see updated Figure 3 below). Specifically, this new data allowed the identification of 10 new strain sharing events between mother-infant pairs, in addition to confirming the two previously reported events. This is now shown in the additional trees included in the updated version of Figure 3D and in the new Figure 3E. In addition, the new data enabled a more robust description of the microbiome overlap between milk and infant stool samples at the species-level, as shown in the new Figures 3A, 3B and 3C.

Figure 3. (A) Percentage of the infant stool microbiome that is shared with the respective mother, across all mother-infant pairs. Species sharing between milk samples at one month postpartum and infant stool samples at one month and six months of age are shown on the left and right boxplot, respectively. T-test, * for $p \leq 0.05$. Only mother-infant pairs for which milk at one month and stool samples at one and six months were available were included ($n=119$). (B) Overview of the ten most frequently shared species between the infant stool samples at one month of age and the respective mother's milk samples at one month postpartum. *BL. infantis* = *B. longum* subsp. *infantis*; *BL. longum* = *B. longum* subsp. *longum*. (C) Mean percentage of mother-infant

pairs in species belonging to the *Bifidobacterium* genus were shared between the milk and infant stool samples (values reported in grey), were found only in the milk samples or were found only in the infant stool samples. Species sharing is calculated comparing milk at one month versus infant stool samples at one and six months, and milk at 3 months versus infant stool samples at 6 months. Only mother-infant couples for which at least one milk and one infant stool sample was collected are included. (D) Examples of strain sharing between milk and infant stool samples for *B. longum* (SGB17248), *Phocaeicola vulgatus* (SGB1814), *Streptococcus salivarius* (SGB8005) and *Klebsiella pneumoniae* (SGB10115), highlighted in black. Instances of strains found within the same infant at both one and six months of age (persistent strains) are highlighted with grey. (E) Percentage of strains shared between mother-infant pairs divided by infant stool collection time point. (F) Taxonomic composition of strains found to be persistent in the infant gut over time (until six months of age). (G) Percentage of strains in infant stool at one month that persisted at six months divided by delivery mode. For each infant, persistence was calculated as the number of strains shared between the one- and six-month timepoints, divided by the total number of strains detected at one month. Each dot is a mother-infant pair. P-value calculated with t-test, * for $p \leq 0.05$.

We agree with the reviewer that read depth is an important factor in profiling strain sharing. Indeed, we find a correlation between read depth and strain sharing events; namely, the samples for which strain sharing events were identified were also the samples for which we had a higher number of microbial reads from milk samples mapping to MetaPhlan4 database. We have expanded the discussion section of the manuscript to highlight this fact:

“The low microbial load in breast milk samples limited our ability to detect strain sharing between milk and the infant gut, particularly in the milk samples collected at three months, suggesting that the observed sharing patterns likely represent an underestimate. Notably, the samples with the higher number of strain sharing events corresponded to mother-infant pairs 1 and 173, which ranked first and third, respectively, in the number of reads mapped to MetaPhlan4 marker gene database. These findings suggest that higher microbial yield and deeper sequencing depth for milk samples should be a priority for future metagenomic studies.”

The starting volume of breast milk used in this study was 1 ml. This information is now included in the Methods:

“Starting volume for extraction was 1 ml for breast milk, and the weight comparable to a pea-sized scoop for the infant feces.”

While the starting volume of 1 ml and the elution volume of 100ul are generally in line with other recently published metagenomic studies on human milk microbiome^{1,3}, we agree that a higher starting volume would likely improve microbial yield and should be considered in future studies. We have included a sentence in the Discussion section to highlight the need for a standardized

and optimized protocol for the study of human milk microbiome, to guide researchers in the choice of optimal parameters, such as sample's starting volume:

“The definition of a standardized and optimized protocol for human breast milk extraction to maximize microbial yield should also be a priority in the field”.

Given the potential issues with the methods, it is important to note the current results are in disagreement with all other published milk microbiome studies. There are several published milk microbiome studies now that all show dominance of Bacillota (Firmicutes) phylum, specifically Staphylococcus and/or Streptococcus. In contrast, the current study shows the dominance of Bifidobacterium species in milk (and infant stool), which is considered a gut commensal. Considering this study sequenced both milk and stool, the latter being expected to have high Bifidobacterium, I have concerns there may have been sample cross-contamination e.g., during extractions, library prep, or sequencing bleedover. This needs to be checked, for instance, the authors could re-extract a random selection of milk samples and perform metagenomics and 16S rRNA gene sequencing. Metagenomics will confirm the results of the current study and 16S rRNA gene sequencing will demonstrate if the lack of concordance with previous literature is due to the sequencing method (as most previous milk microbiome studies used amplicon-based sequencing).

Response 1.3. We thank the Reviewer for raising these very important points that we did not discuss in the original manuscript. We agree that many previous studies on the milk microbiome are indeed dominated by *Staphylococcus* and *Streptococcus*⁴⁻⁸. In our study, we found both these genera to be present at lower relative abundances than the mean relative abundance reported in the literature (*Staphylococcus*: 6% in our study vs 35.8% reported in Ruan et al.⁶; *Streptococcus*: 3% in our study vs ~20% reported in Ruan et al.⁶). Additionally, our study identified bifidobacteria as the most common genus in human milk samples. Although bifidobacteria are commonly regarded as gut commensals in infants⁹, their presence in human milk samples has been reported by several previous metagenomic studies^{3,8,10}. For example, bifidobacteria were found in breast milk at 28% abundance by Kordy et al⁸, ~15% abundance by Zhang et al¹⁰, and 24% abundance in our study. In addition, Feehily et al.³ have confirmed the presence of bifidobacteria using cultivation-based methods³, isolating multiple species, including *B. breve*, *B. longum*, *B. pseudocatenulatum*, *B. infantis* and *B. bifidum*, from human breast milk. Similarly, Collado et al.¹¹ and Martin et al.¹² detected bifidobacteria in milk from healthy women using quantitative real-time PCR-based approaches. Supporting these findings functionally, Lou et al. showed that *B. breve* was the only infant gut species capable of growing in a human milk-derived medium¹³. Considering these findings altogether, we believe our study is not an outlier, and the composition of the microbiome in our study is not in disagreement with the field.

While we believe these studies provide sufficient evidence to support the presence of bifidobacteria in our milk samples, we acknowledge that our results are in disagreement with the studies that identified *Staphylococcus* and *Streptococcus* as most common taxa in human milk^{4,7,8}. Importantly, these studies were all amplicon-based, while our study used shotgun metagenomic sequencing. The discrepancy may stem from technical differences between amplicon and shotgun metagenomic approaches, as was previously reported in other sample types^{14,15}. Amplicon sequencing, for instance, is particularly susceptible to amplification biases¹⁶. Notably, bifidobacteria are often under-represented in infant stool samples when using this method¹⁵. In fact, universal PCR primers that are not specifically optimised for the detection of bifidobacteria are ill-equipped to detect bifidobacteria even in samples with high microbial mass such as infant stool samples¹⁷. In a direct comparison of sequencing methods, Kordy et al. found that the relative abundance of *B. breve* was significantly higher in metagenomic data compared to the *Bifidobacterium* genus detected via amplicon sequencing⁸. Conversely, some genera, such as *Staphylococcus*, may be over-represented in amplicon data, depending on the choice of the targeted hypervariable region¹⁸. An additional factor contributing to this discrepancy may be the inclusion and duration of the lysis step using bead beating during DNA extraction. Previous studies have shown that the recovery of Bifidobacteria is significantly influenced by whether bead beating is included in the extraction protocol^{19,20} and by the duration of this step²¹. The extended lysis step used in our study (10 minutes) has been reported to enhance the recovery of bifidobacteria compared to shorter durations²¹.

Although a comprehensive assessment of the impact of sequencing approach or lysis duration on the human milk microbiome composition is beyond the scope of this article, we have included a summary of these points in the discussion. Finally, we specify that milk and infant stool samples were processed separately, with sample handling, DNA extraction, and library prep performed independently, minimising the risk of cross-contamination.

“The bifidobacteria signature identified in our milk samples contrasts with some previous amplicon-based studies, which have typically reported the milk microbiome as being dominated by Staphylococcus and/or Streptococcus^{4-7,22}. While bifidobacteria are widely recognized as infant gut commensals⁹, their presence in human milk has also been documented in prior metagenomic studies^{3,10,22} and confirmed through cultivation-based³ and quantitative real-time PCR approaches^{11,12}. This discrepancy between Bifidobacterium- and Staphylococcus/Streptococcus-dominated profiles may stem from amplification biases inherent to amplicon sequencing, as reported in other sample types^{14,15}, leading to the under-representation of bifidobacteria¹⁷ and over-representation of Staphylococcus¹⁸. Notably, the use of universal PCR primers not specifically optimized for bifidobacteria can lead to their under-detection¹⁷. Furthermore, direct comparison of sequencing methods showed higher relative abundances of bifidobacteria in shotgun metagenomic data compared to amplicon-based data²². Additional contributing factors may include variation in DNA extraction protocols²⁰, particularly the inclusion and extended duration (10 min) of the bead-beating lysis step, which has been shown to affect the detection and relative abundance of bifidobacteria²¹. In our study, DNA extractions for milk and infant stool samples were conducted independently

by experienced personnel, therefore minimizing the likelihood of cross-contamination during this step.”

In addition, we have included in the Methods the duration of the lysis:

“Cell lysis was carried out by bead-beating for 10 minutes, followed by elution with 100 µl of the provided elution solution.”

That the authors could not link the transmission of *Bifidobacterium* strains from milk to the infant is a weakness of the current study. Given the shortcomings of the metagenomic approach, could they culture *Bifidobacterium* from both sample types and confirm this with WGS?

Response 1.4. We agree that the lack of sharing of *Bifidobacterium* was a weakness of the previous version of the paper. We address this by generating deeper sequencing data (30 million reads) from a subset of 30 milk samples. This new data allowed us to detect 10 additional strain sharing events, in addition to the 2 initially identified. In particular, we have identified both *B. longum* and *B. bifidum* among the shared strains. We have updated the main text and **Figure 3D** as follows:

*“We identified twelve instances of strain-sharing between a mother’s milk and her infant’s stools, spanning across six mother-infant pairs and ten different taxa: *B. longum* (SGB17248), *B. bifidum* (SGB17256), *P. vulgatus* (SGB1814), *E. coli* (SGB10068), *K. pneumoniae* (SGB10115), *K. variicola* (SGB10114), *V. parvula* (SGB6939), *R. mucilaginosa* (SGB16985), *S. salivarius* (SGB8005), and a strain belonging to the *Streptococcus* genus (SGB8007_group, Fig. 3D and Suppl. Table 4).“*

Figure 3. (D) Examples of strain sharing between milk and infant stool samples for *B. longum* (SGB17248), *Phocaeicola vulgatus* (SGB1814), *Streptococcus salivarius* (SGB8005) and *Klebsiella pneumoniae* (SGB10115), highlighted in black. Instances of strains found within the same infant at both one and six months of age (persistent strains) are highlighted with gray.

Related to the above, the high abundance of a strict anaerobe like Bifidobacterium has not been reported in milk and is unexpected since milk is not anaerobic. Thus, proving the ability to culture Bifidobacterium from milk would alleviate some concerns with the unusual finding that Bifidobacterium species dominate human milk.

Response 1.5. We thank the Reviewer for raising this important point. As discussed above in Response 1.3 and 1.4, several previous studies using metagenomics have shown that bifidobacteria are abundant in human breast milk and that bifidobacteria can be cultivated from

milk samples^{1,3}. Therefore, the detection of bifidobacteria in the breast milk samples included in this study is not new or surprising.

We agree with the reviewer that the presence of bifidobacteria described in this and other recent milk metagenomic studies is not in line with the results of amplicon-based studies^{4,7}. As we discussed in Response 1.3, this discrepancy might be due to technical differences between amplicon and metagenomic sequencing, which have been previously reported in other sample types. We have included the following paragraph in the Discussion section:

“The bifidobacteria signature identified in our milk samples contrasts with some previous amplicon-based studies, which have typically reported the milk microbiome as being dominated by Staphylococcus and/or Streptococcus^{4-7,22}. While bifidobacteria are widely recognized as infant gut commensals⁹, their presence in human milk has also been documented in prior metagenomic studies^{3,10,22} and confirmed through cultivation-based³ and quantitative real-time PCR approaches^{11,12}. This discrepancy between Bifidobacterium- and Staphylococcus/Streptococcus-dominated profiles may stem from amplification biases inherent to amplicon sequencing, as reported in other sample types^{14,15}, leading to the under-representation of bifidobacteria¹⁷ and over-representation of Staphylococcus¹⁸. Notably, the use of universal PCR primers not specifically optimized for bifidobacteria can lead to their under-detection¹⁷. Furthermore, direct comparison of sequencing methods showed higher relative abundances of bifidobacteria in shotgun metagenomic data compared to amplicon-based data²². Additional contributing factors may include variation in DNA extraction protocols²⁰, particularly the inclusion and extended duration (10 min) of the bead-beating lysis step, which has been shown to affect the detection and relative abundance of bifidobacteria²¹. In our study, DNA extractions for milk and infant stool samples were conducted independently by experienced personnel, therefore minimizing the likelihood of cross-contamination during this step.”

In addition, we have added to the Discussion section a sentence in which we present possible mechanisms by which *Bifidobacterium*'s presence in breast milk could be explained:

“Beyond methodological considerations, the biological presence of bifidobacteria in breast milk is also supported by the ability of B. longum and other bifidobacteria to survive and grow at low-oxygen conditions²³⁻²⁶. In addition, oxygen-utilizing taxa commonly found in breast milk, such as Streptococcus and Staphylococcus, may reduce oxygen concentrations, facilitating the growth of bifidobacteria²⁷.”

Mother-infant pairs 1 and 173 were the only pairs found to have strain sharing events, and subsequent analysis found these pairs were most related in shared function and

ARGs. Were these milk samples among the deepest sequenced / had the highest number of reads mapping to the marker gene database?

Response 1.6. This is an important point. We note that since we have performed additional deep sequencing in the revision, these results have changed in the updated manuscript: We have identified a total of 12 strain sharing events across 6 mother-infant pairs (Pair 1, 173, 97, 144, 125 and 53). These pairs ranked respectively, 3rd, 20th, 7th, 11th, 17th and 36th place based on the total number of reads (which ranged from 2.3 million to 322,376 reads); and 1st, 3rd, 8th, 14th, 15th and 38th place based on the number of reads mapping to MetaPhlAn4 database (which ranged from 1.4 million to 9,979 reads). Indeed, some of the samples in which strain and ARG sharing events were detected belonged to mother-infant pairs whose milk samples at one month postpartum had some of the highest numbers of reads mapping to the MetaPhlAn4 marker gene database. This is the case for Mother-Infant Pair 1 (in which 5 strain sharing events were detected), which ranked first with 1.4M mapped reads, and Mother-Infant Pair 173 (in which 2 events were detected), which ranked third with 334,789 reads mapped. We have now included this information in the Discussion section, where we address the limitations of the study and acknowledge that higher sequencing depth may have facilitated the detection of shared strains and ARGs in these particular samples:

“We identified twelve instances of strain-sharing between a mother’s milk and her infant’s stools, spanning across six mother-infant pairs and ten different taxa”

“The low microbial load in breast milk samples limited our ability to detect strain sharing between milk and the infant gut, particularly in the milk samples collected at three months, suggesting that the observed sharing patterns likely represent an underestimate. Notably, the samples with the higher number of strain sharing events corresponded to mother-infant pairs 1 and 173, which ranked first and third, respectively, in the number of reads mapped to MetaPhlAn4 marker gene database. These findings suggest that higher microbial yield and deeper sequencing depth for milk samples should be a priority for future metagenomic studies.”

The metadata is not very comprehensive or granular. For example, much of the analysis focuses on breastfeeding at month 1 and 6, defined as exclusive or mixed. However, we know from previous studies the amount of human milk in the diet, whether the milk was expressed or not, what % of the overall diet was human vs bovine milk, etc impact infant gut microbiome and Bifidobacterium. Without this extra granularity in the metadata, the current results of infant gut microbiome largely support other similar studies, and the current study has a smaller sample size than many recent publications.

Response 1.7. We agree with the reviewer that the metadata in the previous version of the paper was lacking, and have now expanded the metadata substantially. We have now included in Supplementary Table 2 information on whether or not the infants had been introduced to

solid foods at 6 months of age (Supplementary Table 2, “Solid_food_6M”). A preview of the updated structure of Supplementary Table 2 is provided below:

Full_name	partial_name	Sample_ID	Sample_type	Mother_Infant_Pair	Post_filtering_QC	Maternal_age	Maternal_bmi	Infant_sex	Delivery_mor	Infant_exclus	Solid_food_6M	Abx_exposur	Prenatal_abx
4M30015MILK_S389	4M30015MILK	S389	Milk_1M	Mother_Infant_Pair_1	Keep			MALE	SVD		No	Yes	
4M60004MILK_S392	4M60004MILK	S392	Milk_1M	Mother_Infant_Pair_2	Keep	30	20.19	FEMALE	CS		No		No
4M60005MILK_S395	4M60005MILK	S395	Milk_1M	Mother_Infant_Pair_3	Discard	30	28.16	FEMALE	CS		No		No
4M60006MILK_S398	4M60006MILK	S398	Milk_1M	Mother_Infant_Pair_4	Keep	35	31.42	FEMALE	CS		No		No
4M60008MILK_S496	4M60008MILK	S496	Milk_1M	Mother_Infant_Pair_5	Keep	31	32.94	FEMALE	SVD		No		No
4M60009MILK_S406	4M60009MILK	S406	Milk_1M	Mother_Infant_Pair_6	Keep	34	36.55	MALE	CS		No		No
4M60012MILK_S409	4M60012MILK	S409	Milk_1M	Mother_Infant_Pair_7	Keep	22	34.67	FEMALE	SVD		No		No
4M60013MILK_S413	4M60013MILK	S413	Milk_1M	Mother_Infant_Pair_8	Keep	33	32.55	MALE	CS		No		No
4M60014MILK_S417	4M60014MILK	S417	Milk_1M	Mother_Infant_Pair_9	Discard	38	30.37	MALE	CS		No		No
4M60017MILK_S420	4M60017MILK	S420	Milk_1M	Mother_Infant_Pair_10	Discard	34	20.31	MALE	SVD		No		No
4M60018MILK_S423	4M60018MILK	S423	Milk_1M	Mother_Infant_Pair_11	Discard	36	19.97	FEMALE	SVD		No		No
4M60020MILK_S430	4M60020MILK	S430	Milk_1M	Mother_Infant_Pair_13	Keep	40	36.93	MALE	SVD		No		No

Supplementary Table 2. Samples metadata. Legend: SVD = Spontaneous Vaginal Delivery; AVD = Assisted Vaginal Delivery; CS = C-section; “Prenatal_abx” are defined as maternal antibiotic intake before the beginning of active labor.

We have also added an additional analysis to check which taxa were present at higher or lower abundance in infants that were still exclusively breastfed at six months versus those that were introduced to solid foods. We identified no statistically significant trends, after performing multiple test correction (FDR = 0.05). We have included this consideration in the Results section:

“We then compared the relative abundance of taxa in the gut of infants introduced to solid foods and those exclusively breastfed at six months of age. We found no taxa with statistically significant differential abundance between infants who were exclusively breastfed at six months and those who were introduced to solid foods (FDR>0.05).”

In addition, we checked whether the introduction of solid food was associated with changes in the ARGs carriage in infant stool samples at six months of age. We did not find a statistically significant difference. These results have been added as reported in the Results section of the manuscript and in the updated Extended Data 14D-F reported below:

“We found no significant difference in the resistome of infants that were born via C-section compared to those born via vaginal delivery ($p=0.34$ and $p=0.97$ at 1 month and 6 months, respectively, t -test), between those exposed to antibiotics and those that were antibiotics-naive ($p=0.80$ and $p=0.14$ at one month and six months, respectively, t -test), those that were introduced to solid foods at six months of age (0.71 , t -test), nor between those exclusively breastfed at six months of age and those fed with a mixture of breast milk and formula ($p=0.35$, t -test; Extended Data 14D-F).”

Extended Data 14. (C) Number of distinct ARG classes identified across body sites and sampling time point. (D-F) Number of ARG genes identified in infant stool samples, divided by delivery mode, history of antibiotic intake and exclusive breastfeeding at 6 months of age, respectively. P-values calculated using t-test, **** for $p \leq 0.0001$, ns for non significant.

Regarding breast milk expression, all breast milk samples were obtained via the use of hospital-grade electric breast pumps, therefore no manual expression was involved. This is now added in the Methods:

“Milk collection was performed as follows: mothers fed their infant from one or both of their breasts, until the infant was satisfied. After two hours, the milk was collected from the right breast using a hospital grade electric breast pump (Medela Symphony; Medela, Inc., Zug, Switzerland), until cessation of production.”

“Milk sampling was performed via the use of breast pumps”.

Finally, we would like to highlight that, while we agree with the Reviewer that this study has a smaller number of infant stool samples compared to some previously published cohorts, our cohort represents, to the best of our knowledge, the biggest collection of metagenomic data from human breast milk, and one of the very few metagenomic studies in which maternal milk data was longitudinally paired with infant stool data.

Minor comments

- Introduction lines 63+: Another reason the milk microbiome is understudied compared to e.g., the gut, is because using such samples for research (and therefore reducing the volume received by the infant) has important implications which are not relevant when studying e.g., a stool sample.

Response 1.8. We agree with the Reviewer and have added this important point to the introduction:

“This is especially true for colostrum, the milk produced in the first days postpartum, as its particularly low production volume further exacerbates the low biomass challenge typical of all milk samples⁵, and its collection would reduce the amount available for the infant.”

- It needs to be clear in the first lines of the results where the mothers/infants were recruited from geographically.

Response 1.10. We have added the geographical area from which the mothers were recruited in the first sentence of the Results section:

“We collected and sequenced 507 microbiome samples from 195 mother-infant pairs recruited in Minneapolis, USA.”

In addition, a more detailed explanation of the recruitment process is provided in the Methods section:

- *“The Institutional Review Boards of the University of Oklahoma, the University of Minnesota, and the Health Partners Institute approved this study (STUDY00009021).”*
- *“Ethnicity and race of the participants are included in Supplementary Table 2.”*
- *“Samples were collected between 2014 and 2023.”*

- The n needs to be clearer in the figure legends. E.g., Figure 2 since panel B will include only subjects with both samples, the n will not be all mothers, whereas in panel C I suspect all mothers are included. The exact n in each analysis should be in the figure and/or legend.

Response 1.11. We have added the number of samples or mother-infant pairs included in the analysis to the following captions:

- Figure 1B: *“Samples included: 122 for milk at one month, 25 for milk three months, 175 for stool one month and 159 for stool at six months of age.”*

- Figure 2B: “Numbers of samples included: $n=145$ for milk at one month, $n=25$ for milk at three months, $n=175$ for stool samples at one month and $n=159$ for stool samples at six months of age.”
- Figure 3A: “Only mother-infant pairs for which milk at one month and stool samples at one and six months were available were included ($n=119$).”

- **Line 119: use of “strains” is not appropriate. Suggest species.**

Response 1.12. We agree, and have carefully reviewed the entire manuscript to ensure that the term is used appropriately and consistently throughout.

- **Lines 141-162: The majority of this section is describing Fig 2B with words, but the figure clearly shows the %s and is easy to follow, so the text does not add anything. It also distracts from the actual analysis that follows.**

Response 1.13. We appreciate the Reviewer’s comment on the clarity of Figure 2B and we agree that its associated description in the main text can be shortened. We shortened the text as follows:

*“Mothers whose breast milk was dominated by *B. longum* (in particular *BL. infantis*) exhibited the most stable milk microbiome over time. Specifically, 64.3% of milk samples in which *B. longum* was the predominant species at one month postpartum maintained this dominance at three months (Fig. 2B and Extended Data 7B). In contrast, none of the milk samples initially dominated by *B. breve* retained that predominance at the three-month time point (Fig. 2B). In infant stool samples, the percentage of samples dominated by *B. longum* increased over time, from 15.7% at one month to 46.6% at six months of age (Fig. 2B). Conversely, the percentage of samples dominated by non-bifidobacteria decreased from 68.5% to 28.1% (Fig. 2B).”*

- **Line 158: “particularly pronounced reduction” – was this significant?**

Response 1.14. The sentence was edited in the revised version of the manuscript to clarify that the prevalence of bifidobacteria increased from one and six months of age, when not stratifying by breastfeeding status at six months. We have added the statistical significance as follows:

*“Overall, the prevalence of bifidobacteria in the infant gut increased from one to six months of age, particularly for those infants that were still exclusively breastfed at six months (Bonferroni adjusted $p=0.08$ for *B. longum*, $p=2.4 \times 10^{-4}$ for *B. breve*, and $p=1.8 \times 10^{-3}$ for *B. bifidum*, Fisher’s exact test; Fig. 2C).”*

- **Line 163-176: it could be clearer this is stool microbiome**

Response 1.15. We have included the clarification at the beginning of the paragraph:

“Next, we looked at the longitudinal stability of the infant gut microbiome composition at the species-level.”

- Extended data 1 is difficult to follow and would be better summarised as a table.

Response 1.16. We agree with the Reviewer that a tabular format for this figure would enhance its clarity for the reader. As requested, we have included the cohort composition information in Supplementary Table 1 (see preview below), which complements the visual representation provided in Supplementary Figure 1. Additionally, we have updated the numbering of the other supplementary tables accordingly.

	type	number of samples	percentage
Delivery mode	Vaginal	110	75.3%%
	Cesarean	35	24%
	Unknow	1	0.7%%
Infant sex	Male	86	49.10%
	Female	89	50.90%
Excl. BF at 6M	Yes	121	76.10%
	No	34	21.40%
	Unknow	4	2.50%
Abx intake after birth (1M)	Yes	32	18.30%
	No	110	62.90%
	Unknow	33	18.80%
Abx intake after birth (6M)	Yes	46	28.90%
	No	92	57.90%
	Unknow	21	13.20%
Perinatal abx intake (1M)	Yes	39	22.30%
	No	134	76.60%
	Unknow	2	1.10%
Perinatal abx intake (6M)	Yes	32	20%
	No	124	78%
	Unknow	3	2%

Supplementary Table 1. Cohort composition

- Considering the large % of host reads in the milk dataset, could the authors use this to determine secretor status of mother?

Response 1.17. This is an excellent idea, and the secretor status of the mother can add an interesting perspective to our results. To address this, we leveraged data that was generated as part of a previous publication, where we determined secretor status based on the concentration of 2'-fucosyllactose in the same breast milk samples²⁸. Using these data, we tested whether the secretor status of the mother was associated with differences in the milk

microbiome richness and composition at 1 month of age. We have excluded milk samples at 3 months of age as the sample size was too small once the samples were stratified by secretor status. We found no statistically significant trend between secretor status of the mother and microbial richness at 1 month postpartum ($p=0.76$; paired t-test). In addition, we found no significant difference in the overall composition of maternal breast milk obtained from secretor versus non secretor mothers.

We have included the maternal secretor status in the metadata table provided in the supplementary materials. In addition, we have added the results on the impact of maternal secretor status on the milk microbiome diversity and composition at 1 month postpartum as Extended Data 2 (see below). We have updated the numbers of the other supplementary figures accordingly. The updated main text and the new Extended Data 2 are reported below:

- Results section: “*Maternal secretor status was not significantly associated with differences in the milk microbiome alpha and beta diversity ($p=0.76$ paired t-test; Extended Data 2A-B).*”

Extended Data 2. (A) Alpha and (B) beta diversity of milk microbiome samples collected at 1 month postpartum stratified by maternal secretor status. See Methods for definition of secretor vs non secretor status. P-value calculated using paired t-test. Ordination plot based on Bray-Curtis distance between samples, colored by maternal secretor status. Milk samples collected at 3 months were not included.

- Results section: “*Exclusive breastfeeding at six months of age, antibiotic intake, and maternal secretor status were not significantly associated with differences in strain persistence in the infant gut ($p=0.08$, $p=0.25$, and $p=0.84$, respectively).*”

- Methods section: “Mothers with an inactive fucosyltransferase 2 (FUT2) gene are classified as non-secretors. Secretor status for the mothers in this study was obtained from²⁸ and was calculated based on the concentration of 2'-fucosyllactose in breast milk”

- Line 334: Bifidobacterium are not oral bacteria. This is related to my earlier comment that skin/oral facultative such as Staphylococcus and/or Streptococcus are expected in human milk, not gut bacteria like Bifidobacterium

Response 1.18. We appreciate the Reviewer for pointing out this issue. Our initial sentence was not intended to refer to Bifidobacteria, but was referring to the presence of typical oral bacteria, such as *S. salivarius*, in the following paragraph. We edited the text to clarify this. The edited sentence now reads:

“The presence in the milk of typical oral species, such as S. salivarius, is likely due to the possible transfer of microbes from the oral cavity of the infant to the breast milk during suckling, a process called retrograde flow²⁹.”

Reviewer #2 (Remarks to the Author):

In this study, Allert *et al.* dive into the connection between the breast milk microbiome and the infant gut microbiome. The authors show that Bifidobacterium, and specifically *B. longum*, overlaps between milk and infant stool samples. Next, they find two strain sharing events between an infant's gut and its mothers breast milk, and show that these dyads also have a high percentage of ARG sharing. The study overall suggests a connection between breast milk microbiome and the influence it has on the infant gut microbiome.

We thank the Reviewer for their thorough and insightful review of our work.

Major comments:

-The introduction is somewhat lacking, specifically regarding previous literature on the breast milk microbiome and on Bifidobacterium species:

a. The authors mention that most previous research on the breast milk microbiome used 16S-rRNA sequencing, however what is found in the current literature should be elaborated on. For example, does previous work show a high abundance of

Bifidobacteria in breast milk? Do the results that are found in this study match what is already known?

Response 2.1. We thank the Reviewer for raising this point and providing the opportunity to better contextualize our study. We have extended the introduction to include the core microbiome signature identified by previous amplicon studies:

“Previous amplicon studies have identified a “core” microbial signature of human breast milk microbiome, mostly dominated by taxa belonging to the Staphylococcus and Streptococcus genera, followed by other genera such as Lactobacillus, Bifidobacterium, Veillonella and Escherichia among others^{27,30}. ”

In addition, we have included a new paragraph where we discuss how the bifidobacteria-dominated signature we identified in this study compares to the *Streptococcus/Staphylococcus*-dominated profiles reported in previously published studies. For a detailed response see Response 1.3; briefly, we note that our study found lower relative abundances of *Staphylococcus* and *Streptococcus* than previous amplicon studies, while identifying Bifidobacteria as the most prevalent genus in human milk. We hypothesize that this difference is likely due to methodological factors (e.g. amplification biases in amplicon versus metagenomic sequencing), resulting in higher recovery rate of bifidobacteria. The new text added to the Discussion reads as follows:

“The bifidobacteria signature identified in our milk samples contrasts with some previous amplicon-based studies, which have typically reported the milk microbiome as being dominated by Staphylococcus and/or Streptococcus^{4-7,22}. While bifidobacteria are widely recognized as infant gut commensals⁹, their presence in human milk has also been documented in prior metagenomic studies^{3,10,22} and confirmed through cultivation-based³ and quantitative real-time PCR approaches^{11,12}. This discrepancy between Bifidobacterium- and Staphylococcus/Streptococcus-dominated profiles may stem from amplification biases inherent to amplicon sequencing, as reported in other sample types^{14,15}, leading to the under-representation of bifidobacteria¹⁷ and over-representation of Staphylococcus¹⁸. Notably, the use of universal PCR primers not specifically optimized for bifidobacteria can lead to their under-detection¹⁷. Furthermore, direct comparison of sequencing methods showed higher relative abundances of bifidobacteria in shotgun metagenomic data compared to amplicon-based data²². Additional contributing factors may include variation in DNA extraction protocols²⁰, particularly the inclusion and extended duration (10 min) of the bead-beating lysis step, which has been shown to affect the detection and relative abundance of bifidobacteria²¹. In our study, DNA extractions for milk and infant stool samples were conducted independently by experienced personnel, therefore minimizing the likelihood of cross-contamination during this step.”

b. Bifidobacterium is a common genus found in the infant gut and is known to utilize human milk oligosaccharides (HMOs) along with many additional health benefits to the infant. However, no focus is given in the introduction to Bifidobacterium and specifically to *B. longum* which is one of the focuses of this study.

Response 2.2. We appreciate the Reviewer's comment and we have extended the introduction to include the importance of *B. longum* for the digestion of HMOs:

*"Bifidobacteria are of particular interest due to their critical role in infant health³¹. They are among the most abundant taxa in the gut of breastfed infants, and their depletion has been associated with immune and metabolic disorders³¹⁻³³. Bifidobacteria, in particular *B. longum* subsp. *infantis*, are able to digest HMOs¹. The ability to use HMOs as a carbon source, as well as the genetic mechanisms underlying this ability, varies across members of the *Bifidobacterium* genus³⁴. Notably, the depletion of genes required for the degradation of HMOs in bifidobacteria has been linked to systemic inflammation and immune dysregulation in infants, highlighting the importance of bifidobacteria-mediated HMOs degradation in infant health³⁵.*

- One of the main messages of the paper is that there is a compositional overlap between breast milk and the infant gut, specifically of Bifidobacteria and *B. longum*.

a. Overlap between breast milk and infant stool should be done at the mother-infant pair level as well as the community level.

The overlap shown is only at the community level and not for individual dyads, therefore it is hard to understand if Bifidobacteria species are just abundant in both types of samples or perhaps this might indicate sharing of species between mother and infant. Overlap of Bifidobacteria at the dyad level will allow further insight. For example, in Fig. 2A it is shown that *B. longum* is the predominant and abundant species in infant gut and breast milk samples, however is this true for matched samples from the same dyad? Do infants with abundant *B. longum* in their gut have mothers with abundant *B. longum* in the breast milk?

Response 2.3. We thank the Reviewer for raising this important point and we agree that this additional analysis would be of relevance. To address this point, we have now included a new Supplementary Figure (Extended Data 10, reported below), showing the correlation between the relative abundances of specific bifidobacteria strains in milk and stool samples. In the plot, reported below, each dot is a mother-infant pair (see caption).

Extended Data 10. Correlation between the relative abundance of specific bifidobacteria in milk and infant stool samples, using Pearson correlation. Each dot is a mother-infant pair. Mean relative abundances were calculated for infant stool samples at one and six months of age (x axis) and milk samples at one and three months postpartum (y axis). Only mother-infant pairs for which at least one milk and one infant stool sample were collected and passed preprocessing were included. R coefficients and p-values are reported. Correlation was calculated using Pearson. Bonferroni-adjusted P-values are reported.

This analysis did not reveal any significant overall correlation between the relative abundance of specific *Bifidobacterium* taxa in breast milk and the corresponding infant stool samples. However, we identified several strain-sharing events involving *B. longum* and *B. bifidum* within

individual mother-infant dyads. The observed differences between the dyad- and the community-level analyses suggests that, while breast milk may serve as a reservoir for taxa like *B. longum*, successful bifidobacteria colonization in the infant gut is shaped by other factors (e.g. prior colonization of other taxa in the infant gut or nutrient availability). Thus, bifidobacteria relative abundance in milk does not necessarily predict their abundance in the paired infant, suggesting more complex mechanisms at play beyond the initial mother-to-infant microbial transmission.

We have added this consideration to the Discussion section:

“Although we observed instances of shared Bifidobacterium strains between breast milk and infant stool in specific mother-infant pairs, we did not find statistically significant correlations in bifidobacteria abundance across these sample types at the population level (i.e. across all mother-infant pairs). This discrepancy between individual dyads and population-level trends suggest that, while breast milk may serve as a reservoir for taxa like B. longum, successful colonization in the infant gut is likely influenced by additional factors (i.e., pre-existing microbial community composition, host genetics, or nutrient availability). Consequently, the relative abundance of bifidobacteria in breast milk does not necessarily predict their relative abundance in the infant gut, suggesting more complex mechanisms at play beyond the initial mother-to-infant microbial transmission.”

We have also updated the Results section accordingly:

“At the population level (i.e. across all mother-infant pairs) and after multiple test correction, we found no significant association between the relative abundance of BL. infantis, BL. longum, B. breve, and B. bifidum in breast milk and in the stool samples of the respective infants (Extended Data 10).”

b. Sharing analysis should be added for the species level, rather than just the strain level. The authors elaborate and find two events of strain sharing between breast milk and the infant gut. It is clear that on the strain level it is hard to find sharing events due to the low coverage in the milk samples, however there is no reference to sharing at the species level. Perhaps analyzing shared species and specifically Bifidobacterium species between breastmilk and the infant gut will allow further insight into the possible sharing events.

Response 2.4. We thank the Reviewer for this suggestion and we agree that an overview of the species that were found in both the maternal milk and the infant gut microbiome would be useful to the reader. We have performed a new analysis to show the species shared between mother and infant pairs, which is now included in the new Figure 3A, 3B and 3C:

Figure 3. (A) Percentage of the infant stool microbiome that is shared with the respective mother, across all mother-infant pairs. Species sharing between milk samples at one month postpartum and infant stool samples at one month and six months of age are shown on the left and right boxplot, respectively. T-test, * for $p \leq 0.05$. Only mother-infant pairs for which milk at one month and stool samples at one and six months were available were included ($n=119$). (B) Overview of the ten most frequently shared species between the infant stool samples at one month of age and the respective mother’s milk samples at one month postpartum. *BL. infantis* = *B. longum* subsp. *infantis*; *BL. longum* = *B. longum* subsp. *longum*; (C) Mean percentage of mother-infant pairs in species belonging to the *Bifidobacterium* genus were shared between the milk and infant stool samples (values reported in grey), were found only in the milk samples or were found only in the infant stool samples. Species sharing is calculated comparing milk at one month versus infant stool samples at one and six months, and milk at 3 months versus infant stool samples at 6 months. Only mother-infant couples for which at least one milk and one infant stool sample was collected are included.

The main text has also been expanded as follows:

“While maternal microbial transmission is well documented for other body sites, it remains unclear whether specific taxa are shared between breast milk and the infant gut, which microbes are involved, and how frequently such sharing occurs. To investigate these questions, we quantified species- and strain-level overlap between breast milk and infant stool samples to investigate potential vertical transmission events (see Methods). We found that species sharing between a mother’s milk and her infant’s stool samples decreased over time. On average, 10.1% of taxa found in infant stool samples at one month of age were also found in their respective mother’s milk collected at one month postpartum (Fig. 3A). This percentage significantly decreased when considering infant stool samples at 6 months of age (7.2%, $p=1.5 \times 10^{-2}$ t-test; Fig. 3A). The most commonly shared species between milk and infant stool samples at one month postpartum was *B. longum* (shared on average in 44.3% of mother-infant pairs - 46.9% for *BL. infantis* and 41.7% for *BL. longum*), followed by *S. epidermidis* (38.7%), *B. breve* (26%), and *S. salivarius* (18%, Fig. 3B). When considering stool samples at one and six months combined, *BL. infantis* was shared in 54.4% of the mother-infant pairs, while *BL. longum* was shared in 47.2% of pairs (Fig. 3C). *B. breve* and *B. bifidum* were characterized by a significantly lower rate of sharing (30% and 11%, respectively, Figure 3C).”

We have also modified the caption of Figure 3 accordingly:

*“Figure 3. (A) Percentage of the infant stool microbiome that is shared with the respective mother, across all mother-infant pairs. Species sharing between milk samples at one month postpartum and infant stool samples at one month and six months of age are shown on the left and right boxplot, respectively. T-test, * for $p \leq 0.05$. Only mother-infant pairs for which milk at one month and stool samples at one and six months were available were included (n=119). (B) Overview of the ten most frequently shared species between the infant stool samples at one month of age and the respective mother’s milk samples at one month postpartum. BL. infantis = B. longum subsp. infantis; BL. longum = B. longum subsp. longum; (C) Mean percentage of mother-infant pairs in species belonging to the Bifidobacterium genus were shared between the milk and infant stool samples (values reported in grey), were found only in the milk samples or were found only in the infant stool samples. Species sharing is calculated comparing milk at one month versus infant stool samples at one and six months, and milk at 3 months versus infant stool samples at 6 months. Only mother-infant couples for which at least one milk and one infant stool sample was collected are included.”*

We would also like to highlight that, as part of this revision, we have generated new data by re-sequencing a subset (n=30) of the milk samples at a much higher sequencing depth (30 million reads per sample on average). This allowed us to identify a larger number of strain sharing events (n=12) between the mother’s milk and the respective infant stool samples, compared to the initial version of this manuscript (where n=2 strain sharing events were identified). These improved results are now included in the updated version of Figure 3D (see below) and the associated text:

Figure 3. (D) Examples of strain sharing between milk and infant stool samples for *B. longum* (SGB17248), *Phocaeicola vulgatus* (SGB1814), *Streptococcus salivarius* (SGB8005) and *Klebsiella pneumoniae* (SGB10115), highlighted in black. Instances of strains found within the same infant at both one and six months of age (persistent strains) are highlighted with gray.

“While informative of the potential microbial sharing between mothers and infants, species-level taxonomic profiling is not sufficient to identify transmission events, due to the genetic variability between conspecific strains³⁶. We therefore performed strain-level profiling with StrainPhlAn⁴³⁷, which allowed us to reliably identify strains that are found in pairs of samples, such as a mother and her infant, defined as strain sharing. We reconstructed a total of 77 strains, of which 65 were found only in infant fecal samples and 12 were found in milk samples, due to the lower coverage. We identified twelve instances of strain-sharing between a mother’s milk and her infant’s stools, spanning across six mother-infant pairs and ten different taxa: *B. longum* (SGB17248), *B. bifidum* (SGB17256), *P. vulgatus* (SGB1814), *E. coli*

(SGB10068), *K. pneumoniae* (SGB10115), *K. variicola* (SGB10114), *V. parvula* (SGB6939), *R. mucilaginosa* (SGB16985), *S. salivarius* (SGB8005), and a strain belonging to the *Streptococcus* genus (SGB8007_group, Fig. 3D and Suppl. Table 4).”

c. It is not clear how strain sharing events were defined. For example in Fig. 3A for mother-infant pair 1, how was it determined that the three strains shown are the same strain? What is the distance between the strains?

Response 2.5. We agree that the definition was not clear in the previous version of the paper. We have clarified how strain sharing events were identified in the revised manuscript. Strain sharing events were identified using the dedicated python script provided as an integral part of the latest StrainPhlAn4 release. The script takes in input, for each species-level genome bin, the phylogenetic tree computed by StrainPhlAn4 and the samples metadata. It then outputs the list of sample pairs that share the same strain, using precomputed identity thresholds. These are taxa-specific normalized phylogenetic distance thresholds, calculated as leaf-to-leaf branch lengths normalized by total tree branch length in phylogenetic trees produced by StrainPhlAn. The StrainPhlAn4 phylogenetic trees are built on marker gene alignments on positions with at least 1% variability. These precomputed identity thresholds are provided by the developers of StrainPhlAn4, and this approach was also used by other recently published studies^{38–40}. We have expanded the Methods section to include a more detailed description for this part of the analysis, as follows:

“Strain-level profiling was performed using StrainPhlAn4³⁷, based on single-nucleotide variant calling, using default parameters. Strain-sharing events between samples were identified based on the output of the “strain_transmission.py” script, which is part of the StrainPhlAn4 release. Detailed step-by-step description is available in <https://github.com/biobakery/MetaPhlAn/wiki/Strain-Sharing-Inference>. Briefly, the “strain_transmission.py” script takes as input the phylogenetic tree computed by StrainPhlAn4 for each species-level genome bin (SGB), SGB ID, and metadata table containing information on subjectID, relation (e.g. mother-infant pair), and sampling time point. Precomputed thresholds provided in “VallesColomerM_2022_Nov19_thresholds.tsv” were used when possible, otherwise the default threshold of 0.03 was provided to the script. The script outputs a list of all sample pairs in which strain sharing was detected. The collated output of the “strain_transmission.py” script is available as Supplementary Table 4.”

- This study focuses on *B. longum*, a common infant gut commensal, however it is clearly known that there are two main different subspecies within this clade that are found in infants: *Bifidobacteria longum* subsp. *infantis* and *Bifidobacteria longum* subsp. *longum*. There are clear differences between these subspecies and their ability to function in the infant gut, specifically in the context of HMOs. Therefore I suggest using a method that will allow differentiation between these subspecies, and focus on the subspecies that overlaps between breast milk and the infant gut microbiome. This can

be done using the existing metagenomic data (Ennis et al. 2024, Casaburi et al. 2021), or qPCR with subspecies-specific primers. It will also be interesting to see if *B. infantis* is found in breast milk samples.

Response 2.6. We thank the Reviewer for this valuable suggestion and we agree that results on *B. longum* subspecies represent an important addition to this work. After including the novel data generated for this revision, we have re-analysed the entire dataset using a custom MetaPhlan4 marker genes database that was recently published by Ennis *et al* that allows to distinguish *B. longum subsp. infantis* from *B. longum subsp. longum*¹. Analyzing these data we found that the prevalence and abundance of these two subspecies in the infant gut changed over time. In particular, we found that *BL. longum* was more abundant than *BL. infantis* at one month but not at six months of age. The updated taxonomic composition table including *B. longum* subspecies is available as Supplementary Table 3. The results of the *B. longum* subspecies analysis are now reported in the new Figure 3B and 3C, in the new Extended Data 6A, 6B and 6C and in the new Extended Data 7C and 7D. The main text associated with these figures was also modified accordingly as follows:

- Results: “*In both milk and infant stool samples, B. longum subsp. longum (BL. longum) and B. longum subsp. infantis (BL. infantis) frequently co-existed in the same sample (81.4% of milk and stool samples, 52.1% of milk samples at one month, 83.4% and 95.6% of stool samples at one and six months, respectively; Extended Data 6A-B). BL. infantis was present at significantly higher relative abundance than BL. longum in milk samples at one and three months postpartum ($p=3.2 \times 10^{-2}$ and $p=4 \times 10^{-2}$, respectively; t-test) and in infant stool samples at six months of age ($p=1.5 \times 10^{-6}$; t-test - Extended Data 6C). BL. longum was significantly enriched in infant stool samples at one month of age as compared to BL. infantis ($p=1.6 \times 10^{-3}$; t-test - Extended Data 6C).*”

Extended Data 6. (A) Overview of *Bifidobacterium longum* subspecies in milk and infant stool samples, (B) their prevalence and (C) their relative abundance across sample types and over time. **** for $p \leq 0.0001$, ** for $p \leq 0.01$ and * for $p \leq 0.05$. P-values calculated with paired t-test and adjusted with Bonferroni correction. *BL. infantis* = *B. longum* subsp. *infantis*; *BL. longum* = *B. longum* subsp. *longum*.

- Results: “This trend was confirmed also in *B. longum* subspecies (Bonferroni adjusted $p=3.9 \times 10^{-3}$ for *BL. longum*, $p=1.4 \times 10^{-2}$ for *BL. infantis*, Fisher’s exact test; Extended Data 7C).”

Extended Data 7. (C) Prevalence of *Bifidobacterium* species and subspecies at one and six months of age. Prevalence at six months is further stratified by exclusive breastfeeding status (n=121 for exclusive breastfeeding; n=34 for mixed/formula diet). Samples lacking breastfeeding information at six months were excluded from the stratified subgroups (n=4). Bonferroni adjusted p-values were calculated using Fisher’s exact test.

- Results: “Subspecies analysis showed that *BL. infantis* increased from 3.2% mean relative abundance at one month to 23.8% at six months (Bonferroni adjusted $p=3.6 \times 10^{-17}$, Wilcoxon Rank Sum) and that *BL. longum* increased from 8.7% mean relative abundance at one month to 9.1% at six months (Bonferroni adjusted $p=1.1 \times 10^{-5}$, Wilcoxon Rank Sum) (Extended Data 7D). Overall, *BL. longum* was more abundant than *BL. infantis* at one month but not at six months of age (Bonferroni adjusted $p=4.1 \times 10^{-11}$, Wilcoxon Rank Sum, Extended Data 7D).”

Extended Data 7. (D) Relative abundance of bifidobacteria in infants at one and six months of age. Relative abundances at six months of age are further stratified by exclusive breastfeeding status. Relative abundances were calculated only on samples in which the species was present. P- values calculated using Wilcoxon rank sum, * for $p \leq 0.05$, ** $p \leq 0.01$, *** for $p \leq 0.001$ and **** for $p \leq 0.0001$. Reported p-values are adjusted using Bonferroni method.

- Results: “The most commonly shared species between milk and infant stool samples at one month postpartum was *B. longum* (shared on average in 44.3% of mother-infant pairs - 46.9% for *BL. infantis* and 41.7% for *BL. longum*), followed by *S. epidermidis* (38.7%), *B. breve* (26%), and *S. salivarius* (18%, Figure 3B). When considering stool samples at one and six months combined, *BL. infantis* was shared in 54.4% of the mother-infant pairs, while *BL. longum* was shared in 47.2% of pairs (Figure 3C).”

Figure 3. (C) Mean percentage of mother-infant pairs in species belonging to the *Bifidobacterium* genus were shared between the milk and infant stool samples (values reported in grey), were found only in the milk samples or were found only in the infant stool samples. Species sharing is calculated comparing milk at one month versus infant stool samples at one and six months,

and milk at 3 months versus infant stool samples at 6 months. Only mother-infant couples for which at least one milk and one infant stool sample was collected are included.

- Discussion: “At the subspecies level, we found that both *BL. infantis* and *BL. longum* frequently co-existed in the same sample (in 81.4% of combined milk and infant stool samples), and that both were present at high prevalence (>75%) in infants at one month of age. Surprisingly, *BL. infantis* was already detectable at this early time point, in contrast to a recent study that found *BL. infantis* in infant stool samples only after 10 weeks of age¹. Although that study’s meta-analysis supported the delayed appearance of *BL. infantis* across multiple populations, samples collected from the United States cohort were limited to the first two weeks of life, preventing a direct comparison with our samples collected at one month of age. However, in line with the same study¹, we found that *BL. infantis* was significantly more abundant in infant stool samples at six months compared to one month and compared to *BL. longum* at six months of age.”
- Methods: “Species-level taxonomic profiling was based on marker genes using MetaPhlAn4³⁷, with the following parameters: “--bt2_ps sensitive” text. MetaPhlAn4 infers taxonomic prevalence and abundance by using unique marker genes for 26,970 species-level genome bins³⁷. As MetaPhlAn4 has no marker genes to identify and distinguish *B. longum* subspecies (mainly *BL. infantis* and *BL. longum*), we used a modified version of the standard MetaPhlAn4 database. Details of the modified database can be found in the original publication¹, while the database can be found on GitHub (<https://github.com/yassourlab/MetaPhlAn-B.infantis/>).”

We also analysed the updated taxonomic composition table to address the Reviewer’s question on whether we found *BL. infantis* in breast milk samples. We indeed found *BL. infantis* in both milk samples collected at one and three months postpartum, at significantly higher abundance than *BL. longum*. This result has been added to the main text and in the new Extended Data 6C as reported below:

“*BL. infantis* was present at significantly higher relative abundance than *BL. longum* in milk samples at one and three months postpartum ($p=3.2 \times 10^{-2}$ and $p=4 \times 10^{-2}$, respectively; *t*-test) and in infant stool samples at six months of age ($p=1.5 \times 10^{-6}$; *t*-test - Extended Data 6C). *BL. longum* was significantly enriched in infant stool samples at one month of age as compared to *BL. infantis* ($p=1.6 \times 10^{-3}$; *t*-test - Extended Data 6C).”

Extended Data 6. (C) Overview of *Bifidobacterium longum* subspecies relative abundance across sample types and over time. **** for $p \leq 0.0001$, ** for $p \leq 0.01$ and * for $p \leq 0.05$. P-values calculated with paired t-test and adjusted with Bonferroni correction. *BL. infantis* = *B. longum* subsp. *infantis*; *BL. longum* = *B. longum* subsp. *longum*.

- Throughout the manuscript and in the title it is mentioned how the stability and dynamics in the infant gut microbiome is linked to the bacteria in breast milk, however this statement should be changed as currently, it is not proven in the paper. a. When discussing stability (Fig. 2) the analysis is performed separately for breast milk and the infant gut. The main focus is on the infant gut microbiome with differentiation between infants that were exclusively breastfed at 6 months and those that weren't. How does the milk microbiome influence the stability of the infant gut? Is it only being breastfed or not? b. Are there any direct effects of breast milk microbial composition/ specific bacteria that influence the stability or dynamics in the infant gut microbiome? Was this checked and just not significant?

Response 2.7. We thank the Reviewer for the comments and for the opportunity to clarify the analyses we performed. We have assessed the impact of milk on the infant gut stability in multiple ways. (i) First, we assessed whether exclusive breastfeeding until 6 months of age was associated with differences in infant gut microbiome composition. Results and the updated Figure 2F are reported below:

"Infants that were exclusively breastfed through six months of age showed a significant increase in *B. longum* and *B. breve* relative abundances (BH-adjusted $p=3.6 \times 10^{-9}$, $p= 2.2 \times 10^{-5}$,

respectively, paired t-test; Fig. 2F and Extended Data 9A), and a significant reduction in *C. perfringens* (BH-adjusted $p=1.3 \times 10^{-2}$, paired t-test; Fig. 2G), compared to those who were not exclusively breastfed at six months of age."

Figure 2. (G) Relative abundance of some of the differentially abundant species between one and six months when divided by breastfeeding (BF) at six months. P-values calculated with paired t-test and adjusted with BH correction. **** for $P \leq 0.0001$, *** for $P \leq 0.001$, ** for $P \leq 0.01$ and * for $P \leq 0.05$.

(ii) We next examined whether specific species were associated with increased stability of the infant gut microbiome from 1 to 6 months of age, and whether these species were also present in the mother's milk. We found that *B. longum* predominance was indeed associated with increased stability in the infant gut microbiome and that *B. longum* could be acquired from the mother's milk, as:

- *B. longum* was the most prevalent and among the most abundant species in both milk and infant stool samples (see updated Figure 1C-D below)

Figure 1. Taxonomic composition of (C) the most prevalent and abundant species found in the human breast milk and (D) in the infant gut microbiome samples in relation to sample collection time point, predominance group and other relevant infant metadata. The predominance group identifies the most abundant species in each sample. Ordination plot based on Bray-Curtis distance between samples for PCoA1 and PCoA2 (E) and PCoA1 and PCoA3 (F), colored by body site of origin and sampling time. Percentage of variance explained (PVE) is reported in parentheses.

- One of the twelve instances of strain sharing between maternal milk and the infant stool involved *B. longum* in Mother-Infant Pair 1 (see updated Figure 3D below). The *B. longum* strain shared with milk was detected in the infant's stool at both 1 and 6 months of age.

Figure 3. (D) Examples of strain sharing between milk and infant stool samples for *B. longum* (SGB17248), *Phocaeicola vulgatus* (SGB1814), *Streptococcus salivarius* (SGB8005) and *Klebsiella pneumoniae* (SGB10115), highlighted in black. Instances of strains found within the same infant at both one and six months of age (persistent strains) are highlighted with gray.

Combined, these results suggest that maternally-acquired *B. longum* might play a role in the infant gut microbiome stability over time. The novel data generated in this revision significantly improved our detection of microbial strains in breast milk, as well as strain sharing events between mother-infant pairs. In light of these updated results, we have updated the manuscript title to reflect the heightened focus on microbial and resistance transmission between mother and infant pairs:

“Assembly of the infant gut microbiome and resistome are linked to bacterial strains in mother’s milk”.

Minor comments:

- Fig 1E- Axis titles should be PCo1 and Pco2 (instead of PC1 and PC2).

Response 2.8. We have fixed the axis titles for Figure 1E as suggested by the Reviewer. The updated figure is below:

Figure 1. Ordination plot based on Bray-Curtis distance between samples for PCoA1 and PCoA2 (E) and PCoA1 and PCoA3 (F), colored by body site of origin and sampling time. Percentage of variance explained (PVE) is reported in parentheses.

- Fig 2D- is this figure only for samples that have the specific bacteria? If yes, this should be mentioned in the caption. All samples had *B. longum* at 6 months?

Response 2.9. We thank the Reviewer for the opportunity to clarify this point. For Figure 2D, we considered only the samples i) for which we had metadata on breastfeeding at six months of age, and ii) infants in which the target species (*B. longum*, *B. breve* or *B. bifidum*) was present in the sample. In other words, the mean relative abundances were calculated considering only samples in which the taxon of interest was identified. To clarify this point, we have expanded the caption of Figure 2 as follows:

“Relative abundances were calculated only on samples in which the species were identified.”

- In line 164 “We found that *B. longum* and *E. coli* were more abundant at six months compared to one month”, however in Fig. 2E *E. coli* seems to decrease over time. The same is true for exclusively breastfed infants according to Fig. 2F (line 169).

Response 2.10. We thank the reviewer for spotting this mistake in the previous version of the manuscript. Indeed, *E. coli* abundance decreases over time. We have corrected the paragraph as follows:

*“Next, we looked at the longitudinal stability of the infant gut microbiome composition at the species-level. We found that *B. longum* (in particular *B. infantis*) was more abundant at six months compared to one month (BH-adjusted $p=2.2 \times 10^{-10}$, t-test), while *E. coli* and *V. dispar* decreased over time (BH-adjusted $p=5.3 \times 10^{-3}$ and $p=6.7 \times 10^{-3}$, respectively, t-test; Fig. 2E and Extended Data 7E).”*

- Fig 3C- it would be important to mention what species have shared strains between unrelated infants (or what are the main species with shared strains).

Response 2.11. We agree with the Reviewers that this information could be of interest for the reader. However, upon re-analysis of the dataset following the inclusion of re-sequenced milk samples as suggested by Reviewer #1, we could not confirm our initial results on strain sharing events between unrelated infants. For this reason, we have removed those results from Figure 3, and have instead included a new analysis on the persistence of microbial strains in the infant gut between one and six months of age. In particular, we identified the taxa most commonly found in both timepoints (hence persistent over time) and we found that infants born via C-section were characterized by a lower number of persistent strains than those vaginally born. These results are reported as follows:

*“In addition to mother-infant sharing, strain analysis can also provide information on strain persistence in the infant gut over the first six months of age. We found that 19.4% of the strains identified in infant stool samples at one month persisted in the infant stools until six months of age (Suppl. Table 4). The taxa most commonly found to be persistent in the infant gut included *E. coli* (23.2% of persistence events), *B. longum* (13.4%), *B. fragilis* (8.5%), *B. vulgatus* (8.5%), and *B. bifidum* (8.5%, Suppl. Table 4 and Fig. 3F). Vaginally delivered infants had a higher number of persistent strains than infants that were born via C-section ($p=1.8 \times 10^{-2}$, t-test; Fig. 3G). Exclusive breastfeeding at six months of age, antibiotic intake, and maternal secretor status were not significantly associated with differences in strain persistence in the infant gut ($p=0.08$, $p=0.25$, and $p=0.84$, respectively).”*

And in the new Figures 3F-G (see below):

Figure 3. (F) Taxonomic composition of strains found to be persistent in the infant gut over time (until six months of age). (G) Percentage of strains in infant stool at one month that persisted at six months divided by delivery mode. For each infant, persistence was calculated as the number of strains shared between the one- and six-month timepoints, divided by the total number of strains detected at one month. Each dot is a mother-infant pair. P-value calculated with t-test, * for $p \leq 0.05$.

- Fig. 4D- On one hand it is mentioned that strain sharing events are probably missed due to low coverage in breast milk samples, on the other hand significant correlation between metabolic pathways in breast milk and stool samples were found only in dyads where shared strains were found. Can this lack of significant correlation in most dyads be explained by the low coverage in breast milk samples? Perhaps the correlation in pairs 1 and 173 is due to the specific bacteria shared within the pair (*B. longum* and *K. pneumoniae* respectively)- do these bacteria have a high relative abundance in the relevant samples? In addition, do pairs 1 and 173 have a higher coverage in breast milk samples which can explain this?

Response 2.12. We thank the Reviewer for raising this point, which was also raised by Reviewer #1. We agree that the lack of significant correlation in most dyads is likely due to low coverage in milk samples, as we highlighted in the Discussion section:

“The fact that we were able to reconstruct strains from only ten taxa in our milk samples, and yet all were found to be shared between mother’s milk and infant gut, indicates that our estimate for strain sharing between breast milk and infant gut is likely an underestimate due to the low number of microbial reads in the milk samples.”

As suggested by the Reviewer, we compared the relative abundances of *B. longum* and *K. pneumoniae* in mother-infant pairs 1 and 173. In this analysis we found contrasting trends: in mother-infant pair 1, *B. longum* abundance in milk samples at 1 month postpartum and infant stool at 6 months was comparable (90% and 85%, respectively), while in mother-infant pair 173 the abundance of *B. longum* in these timepoints was very different (3.1% and 75.8%, respectively). These contrasting trends were also found when looking at the relative abundances of *K. pneumoniae* in milk versus stool samples. However, as explained in our Response 1.6., milk samples belonging to mother-infant couples 1 and 173 had indeed the highest number of reads mapping to the MetaPhlAn4 marker genes database. Therefore, a higher number of microbial reads will likely improve not only strain sharing analysis but also the analysis of metabolic pathways shared between mothers and their infants. We have included this comment in the discussion section where we discuss the limitations of this study:

“The low microbial load in breast milk samples limited our ability to detect strain sharing between milk and the infant gut, particularly in the milk samples collected at three months, suggesting that the observed sharing patterns likely represent an underestimate. Notably, the samples with the higher number of strain sharing events corresponded to mother-infant pairs 1 and 173, which ranked first and third, respectively, in the number of reads mapped to MetaPhlAn4 marker gene database. These findings suggest that higher microbial yield and deeper sequencing depth for milk samples should be a priority for future metagenomic studies.”

Additional minor additions and corrections:

We thank the Reviewers for their valuable comments and suggestions. We would like to highlight that during the revision process we considered necessary to include the following additional information and corrections:

- In the Methods, we have specified that “*Samples were collected between 2014 and 2023*”.
- We have included the number of the ethical committee approval “*The Institutional Review Boards of the University of Oklahoma, the University of Minnesota, and the Health Partners Institute approved this study (STUDY00009021)*.”
- While re-running our pipeline with newly generated data, we refined the host read removal step using the latest telomere-to-telomere (T2T) human genome, as recommended for low-biomass samples like breast milk (Gihawi et al. 2023), to minimize host contamination and improve microbial profiling accuracy.
- We have specified that “*The Authors declare no conflict of interest.*”
- We have fixed the instances in which the study name was erroneously referred to as “MILK” instead of “MILk”.

- We have included ethnicity and race information in the Supplementary Table 1.
- We have included a legend for the abbreviations used in the Supplementary Table 2 (study metadata).
- In the Methods, we have corrected the name of the extraction kit used: “PowerSoil Pro kit” instead of “PowerSoil kit”.

References

1. Ennis, D., Shmorak, S., Jantscher-Krenn, E. & Yassour, M. Longitudinal quantification of *Bifidobacterium longum* subsp. *infantis* reveals late colonization in the infant gut independent of maternal milk HMO composition. *Nat. Commun.* **15**, 894 (2024).
2. Spreckels, J. E. *et al.* Analysis of microbial composition and sharing in low-biomass human milk samples: a comparison of DNA isolation and sequencing techniques. *ISME Commun* **3**, 116 (2023).
3. Feehily, C. *et al.* Detailed mapping of *Bifidobacterium* strain transmission from mother to infant via a dual culture-based and metagenomic approach. *Nat. Commun.* **14**, 3015 (2023).
4. Kim, S. Y. & Yi, D. Y. Analysis of the human breast milk microbiome and bacterial extracellular vesicles in healthy mothers. *Exp. Mol. Med.* **52**, 1288–1297 (2020).
5. Williams, J. E. *et al.* Human milk microbial community structure is relatively stable and related to variations in macronutrient and micronutrient intakes in healthy lactating women. *J. Nutr.* **147**, 1739–1748 (2017).
6. Ruan, J.-W. *et al.* The composition of the maternal breastmilk microbiota influences the microbiota network structure during early infancy. *J. Microbiol. Immunol. Infect.* **56**, 1084–1097 (2023).
7. Li, Y. *et al.* The effect of breast milk Microbiota on the composition of infant gut Microbiota: A cohort study. *Nutrients* **14**, 5397 (2022).
8. Kordy, K. *et al.* Contributions to human breast milk microbiome and enteromammary transfer of *Bifidobacterium breve*. *Microbiology* (2019).
9. Turrone, F. *et al.* *Bifidobacteria* and the infant gut: an example of co-evolution and natural selection. *Cell. Mol. Life Sci.* **75**, 103–118 (2018).
10. Zhang, Q. *et al.* Optimization of metagenomic detection method for human breast milk microbiome. *Microbiology* (2024).
11. Collado, M. C., Delgado, S., Maldonado, A. & Rodríguez, J. M. Assessment of the bacterial diversity of breast milk of healthy women by quantitative real-time PCR. *Letf. Appl. Microbiol.* **48**, 523–528 (2009).
12. Martín, R. *et al.* Isolation of bifidobacteria from breast milk and assessment of the bifidobacterial population by PCR-denaturing gradient gel electrophoresis and quantitative real-time PCR. *Appl. Environ. Microbiol.* **75**, 965–969 (2009).
13. Lou, Y. C. *et al.* Infant microbiome cultivation and metagenomic analysis reveal *Bifidobacterium* 2'-fucosyllactose utilization can be facilitated by coexisting species. *Nat. Commun.* **14**, 7417 (2023).
14. Durazzi, F. *et al.* Comparison between 16S rRNA and shotgun sequencing data for the taxonomic characterization of the gut microbiota. *Sci. Rep.* **11**, 3030 (2021).
15. Peterson, D. *et al.* Comparative Analysis of 16S rRNA Gene and Metagenome Sequencing in Pediatric Gut Microbiomes. *Front. Microbiol.* **12**, 670336 (2021).
16. Ferretti, P. *et al.* Experimental metagenomics and ribosomal profiling of the human skin

- microbiome. *Exp. Dermatol.* **26**, 211–219 (2017).
17. Sim, K. *et al.* Improved detection of bifidobacteria with optimised 16S rRNA-gene based pyrosequencing. *PLoS One* **7**, e32543 (2012).
 18. Meisel, J. S. *et al.* Skin Microbiome Surveys Are Strongly Influenced by Experimental Design. *J. Invest. Dermatol.* **136**, 947–956 (2016).
 19. de Boer, R. *et al.* Improved detection of microbial DNA after bead-beating before DNA isolation. *J. Microbiol. Methods* **80**, 209–211 (2010).
 20. Ward, T. L., Hosid, S., Ioshikhes, I. & Altosaar, I. Human milk metagenome: a functional capacity analysis. *BMC Microbiol.* **13**, 116 (2013).
 21. Zhang, B. *et al.* Impact of bead-beating intensity on the genus- and species-level characterization of the gut microbiome using amplicon and complete 16S rRNA gene sequencing. *Front. Cell. Infect. Microbiol.* **11**, 678522 (2021).
 22. Kordy, K. *et al.* Contributions to human breast milk microbiome and enteromammary transfer of *Bifidobacterium breve*. *PLoS One* **15**, e0219633 (2020).
 23. Shimamura, S. *et al.* Relationship between oxygen sensitivity and oxygen metabolism of *Bifidobacterium* species. *J. Dairy Sci.* **75**, 3296–3306 (1992).
 24. Ruiz, L. *et al.* Molecular clues to understand the aerotolerance phenotype of *Bifidobacterium animalis* subsp. *lactis*. *Appl. Environ. Microbiol.* **78**, 644–650 (2012).
 25. Schell, M. A. *et al.* The genome sequence of *Bifidobacterium longum* reflects its adaptation to the human gastrointestinal tract. *Proc. Natl. Acad. Sci. U. S. A.* **99**, 14422–14427 (2002).
 26. Zhang, D. *et al.* Oxygen tolerance mechanism of *Bifidobacterium animalis* AR668-R1 based on genomic and phenotypic analyses. *Lebenson. Wiss. Technol.* **215**, 117207 (2025).
 27. Mantziari, A. & Rautava, S. Factors influencing the microbial composition of human milk. *Semin. Perinatol.* **45**, 151507 (2021).
 28. Johnson, K. E. *et al.* Human milk variation is shaped by maternal genetics and impacts the infant gut microbiome. *bioRxiv* (2023) doi:10.1101/2023.01.24.525211.
 29. Lopez Leyva, L., Brereton, N. J. B. & Koski, K. G. Emerging frontiers in human milk microbiome research and suggested primers for 16S rRNA gene analysis. *Comput. Struct. Biotechnol. J.* **19**, 121–133 (2021).
 30. Notarbartolo, V., Giuffrè, M., Montante, C., Corsello, G. & Carta, M. Composition of Human Breast Milk Microbiota and Its Role in Children's Health. *Pediatr Gastroenterol Hepatol Nutr* **25**, 194–210 (2022).
 31. Saturio, S. *et al.* Role of bifidobacteria on infant health. *Microorganisms* **9**, 2415 (2021).
 32. Fujimura, K. E. *et al.* Neonatal gut microbiota associates with childhood multisensitized atopy and T cell differentiation. *Nat. Med.* **22**, 1187–1191 (2016).
 33. Zhang, S. & Dang, Y. Roles of gut microbiota and metabolites in overweight and obesity of children. *Front. Endocrinol. (Lausanne)* **13**, 994930 (2022).
 34. Asakuma, S. *et al.* Physiology of consumption of human milk oligosaccharides by infant gut-associated bifidobacteria. *J. Biol. Chem.* **286**, 34583–34592 (2011).
 35. Henrick, B. M. *et al.* Bifidobacteria-mediated immune system imprinting early in life. *Cell*

- 184**, 3884–3898.e11 (2021).
36. Van Rossum, T., Ferretti, P., Maistrenko, O. M. & Bork, P. Diversity within species: interpreting strains in microbiomes. *Nat. Rev. Microbiol.* **18**, 491–506 (2020).
 37. Blanco-Míguez, A. *et al.* Extending and improving metagenomic taxonomic profiling with uncharacterized species using MetaPhlAn 4. *Nat. Biotechnol.* (2023)
doi:10.1038/s41587-023-01688-w.
 38. Andreu-Sánchez, S. *et al.* Global genetic diversity of human gut microbiome species is related to geographic location and host health. *Cell* **0**, (2025).
 39. Collado, M. C. *et al.* Birthmode and environment-dependent Microbiota transmission dynamics are complemented by breastfeeding during the first year. *Curr. Dev. Nutr.* **8**, 103460 (2024).
 40. Dubois, L. *et al.* Paternal and induced gut microbiota seeding complement mother-to-infant transmission. *Cell Host Microbe* **32**, 1011–1024.e4 (2024).

Reviewer #1 (Remarks to the Author):

The additional work substantially improves the quality and robustness of the work.

We thank the Reviewer for the positive evaluation of our revised work.

Reviewer #2 (Remarks to the Author):

The authors have addressed most of the main comments. Milk samples were sequenced more deeply, separated between *B. longum* and *B. infantis*, and added relevant points and limitations regarding their study to both the Introduction and Discussion. Figure 3 was refined substantially, including species-level sharing, and deeper sequencing allowed them to identify more strain-sharing results and define strain-sharing events more clearly.

However, since the milk microbiome is of such low biomass, the authors should demonstrate how the samples that were sequenced more deeply match and correlate with their previous results from these same samples, as this would strengthen confidence in the metagenomic analysis of these low-biomass samples.

We thank the Reviewer for their comments on the revised figures and manuscript. To show that the samples sequenced more deeply correlate with the previous results from the same samples, we performed a comparative analysis using the subset of 30 milk samples that were re-sequenced at higher sequencing depth (see figure below). Panel A shows a heatmap of the relative abundances of the top most prevalent species across matched pre- and post-resequencing samples. The heatmap shows consistent presence and relative abundance patterns of the dominant milk species across sequencing runs, indicating that deeper sequencing did not drastically alter the overall microbiome composition. We then calculated the Shannon diversity for each sample before and after re-sequencing (panel B) and observed a strong positive correlation (Spearman's $R = 0.73$, $p = 8.9 \times 10^{-6}$), suggesting alpha diversity metrics were also well preserved between sequencing runs. Finally, in panel C, we examined the species-level reproducibility of the results by plotting the relative abundances of key taxa in matched samples pre- and post-resequencing. These scatterplots show strong correlations for all species ($0.6 < R < 1.0$), with statistically significant p-values across all taxa. Together, these results provide strong support for the technical reproducibility of our results on milk microbiome diversity and composition.

Caption:

Technical reproducibility of milk microbiome profiles before and after re-sequencing. (A) Heatmap showing the taxonomic composition of $n = 30$ milk samples before (pre) and after (post) re-sequencing at higher sequencing depth. Rows represent the top most abundant species, and columns represent matched samples. (B) Correlation of Shannon diversity between pre- and post-resequencing samples. Each point represents a sample; Spearman's correlation coefficient (R) and p -value are shown. (C) Correlation of relative abundances for the most prevalent species across matched samples before and after re-sequencing. Each point represents a sample; dashed lines show linear regression fits. Spearman's R and p -values are reported for each species. Taxa shown include *Bifidobacterium longum*, *B. breve*, *B. bifidum*, *Staphylococcus epidermidis*, *Escherichia coli*, *Cutibacterium acnes*, *Phocaeicola vulgatus*, and *Mycobacterium gordonae*.